# Caspase-1 inhibition alleviates cognitive impairment and neuropathology in an Alzheimer's disease mouse model

Joseph Flores [1,2], Anastasia Noël [1,2], Bénédicte Foveau [1], Jeffrey Lynham [1,3], Clotilde Lecrux [2] & Andréa C. LeBlanc [1,2,3]

Alzheimer's disease (AD) is an intractable progressive neurodegenerative disease characterized by cognitive decline and dementia. An inflammatory neurodegenerative pathway, involving Caspase-1 activation, is associated with human age-dependent cognitive impairment and several classical AD brain pathologies. Here, we show that the nontoxic and blood–brain barrier permeable small molecule Caspase-1 inhibitor VX-765 dose-dependently reverses episodic and spatial memory impairment, and hyperactivity in the J20 mouse model of AD. Cessation of VX-765 results in the reappearance of memory deficits in the mice after 1 month and recommencement of treatment re-establishes normal cognition. VX-765 prevents progressive amyloid beta peptide deposition, reverses brain inflammation, and normalizes synaptophysin protein levels in mouse hippocampus. Consistent with these findings, Caspase-1 null J20 mice are protected from episodic and spatial memory deficits, neuroinflammation and Aβ accumulation. These results provide in vivo proof of concept for Caspase-1 inhibition against AD cognitive deficits and pathologies.

---

[1] Bloomfield Center for Research in Aging, Lady Davis Institute for Medical Research, Jewish General Hospital, 3755 Ch. Côte Ste-Catherine, Montreal, QC H3T 1E2, Canada. [2] Department of Neurology and Neurosurgery, McGill University, 845 Sherbrooke St West, Montreal, QC H3A 0G4, Canada. [3] Department of Anatomy and Cell Biology, McGill University, 845 Sherbrooke St West, Montreal, QC H3A 0G4, Canada. Correspondence and requests for materials should be addressed to A.C.L. (email: andrea.leblanc@mcgill.ca)

Currently, no treatment significantly delays, or reverses AD-associated cognitive decline or pathologies. Targeting earlier degenerative events may be essential to stem the scourge of AD. Preclinical involvement of inflammation in AD, genetic association of immune-related genes with AD, and increased risk of AD in systemic and CNS conditions that increase inflammation support the hypothesis that inflammation is an integral part of AD[1]. In addition, increasing support for the role of inflammation in AD is provided from AD mouse models. AD mouse models on complement 3[2], colony stimulating factor 1 receptor[3,4], inflammasome Nod-like receptor protein 3 (Nlrp3)[5], Caspase-1 (Casp1)[5], scavenger receptor class B type I[6], poly (ADP-ribose) polymerase 1[7], or complement 1qa[8] null background, show improvements in cognitive behaviour, synaptic pathology, and often amyloid beta peptide (Aβ) pathology. Inhibition of inflammasome activator P2X7R decreases Aβ[9], and silencing of inflammasome Nod-like receptor protein 1 (Nlrp1) improves cognition in AD models[10].

We identified a human neurodegenerative pathway, mediated by neuronal Nlrp1 inflammasome activation of Casp1, which then activates Caspase-6 (Casp6), in stressed CNS human primary neuron cultures and AD brains[11]. Active Casp6 is abundant in neuritic plaques, neurofibrillary tangles, and neuropil threads of sporadic and familial AD brains[12,13], and Nlrp1 immunopositive neurons are increased 20−25-fold in AD[11]. Entorhinal cortex and hippocampal CA1 Casp6 activation is associated with age-dependent cognitive impairment in humans[14,15] and in transgenic mice in the absence of other AD pathologies[16]. Active Casp6 is associated with axonal degeneration[17–19], cleaves several cytoskeleton or cytoskeletal-associated proteins including Tau, αTubulin, Drebrin, Spinophillin, Actinin-1 and -4 synaptic proteins[20], increases Aβ production[21–23] and impairs valosin-containing protein p97 proteasome-mediated misfolded protein degradation[24]. Consequently, Nlrp1, Casp1, and Casp6 represent feasible therapeutic targets against age-dependent cognitive deficits and AD.

While there are currently no suitable inhibitors for Nlrp1 and Casp6, VX-765, a pro-drug that is rapidly metabolized in vivo to VRT-043198, is a potent bioavailable, blood–brain barrier permeable, and nontoxic Casp1 small molecule inhibitor[25,26]. VRT-043198 is reported to be selective for Casp1 against Casp3−10, Casp14, granzyme B, trypsin and cathepsin B[26,27]. VX-765 inhibits Il-1β release and Il-18 in activated peripheral blood mononucleated cells and in lipopolysaccharide-induced inflammation, oxalozone-induced dermatitis, and rheumatoid arthritis mouse models[27]. VX-765 blocks HIV-induced pyroptosis of CD4 T cells[28] but lacks significant antiapoptotic activity against hypoxia or Fas-induced apoptosis, eliminating the risk of potential inhibition of physiological apoptotic events in vivo[27]. VX-765 is safe for humans by oral administration as tested in a phase 2b human clinical trial against epilepsy[29]. Therefore, VX-765 represents a safe drug that could rapidly be tested against human age-dependent cognitive impairment and AD.

Here, VX-765 was given to the J20 APP[Sw/Ind] transgenic mouse model[30]. J20 mice show episodic memory deficits by 3−4 months, spatial memory and learning deficits by 6−7 months[31], increased total hippocampal Aβ42 levels at 2 months of age, Aβ plaque formation by 7 months[30], and synaptophysin loss by 2−6 months[30]. Microglial inflammation and neuronal loss precede plaque deposition[32]. The early manifestation of these symptoms and pathologies allows rapid analyses of the VX-765 effects.

## Results

**VX-765 rescues cognitive deficits in symptomatic J20 mice.** VX-765 and VRT-043198 potently and specifically inhibited human Casp1 (IC50 3.68 nM and 9.91 nM) relative to human Casp2−10 (see Supplementary Fig. 1a, b). Similarly, mouse recombinant Casp1 (IC50 52.1 nM and 18 nM) was strongly inhibited compared to inflammatory mouse Casp11 (see Supplementary Fig. 1c, d). Although variable between individual mice, VX-765 crossed the blood–brain barrier of WT and J20 mice, was metabolized into VRT-043198, and reached physiologically active concentrations in both the hippocampus and cortex (see Supplementary Fig. 1e).

Five-month-old mice were behaviourally and longitudinally assessed before treatment (baseline), after 3 injections per week with 50 mg kg$^{-1}$ VX-765 (Treatment 1; T1), after an additional 2 weeks of injections (Treatment 2; T2), after 4 weeks without treatment (Washout; WO), and after 3 more injections per week (Treatment 3; T3) before sacrificing the mice at 8 months of age (Fig. 1a). At baseline, J20 and littermate WT mice showed normal motivation behaviour determined by % time moving (see Supplementary Fig. 2a) and did not exhibit thigmotaxis indicative of anxiety (see Supplementary Fig. 2b), but J20 mice showed a strong deficit in the novel object recognition (NOR) episodic (retention) memory discrimination index (Fig. 1b). J20 NOR deficits were reversed and reached near-normal levels after VX-765 T1 and T2, reappeared after WO, and renormalized after T3. Results were consistent across individual mice (see Supplementary Fig. 2c). J20 mice hyperactivity, measured by distance travelled in the open field task, was attenuated by VX-765 after T2, reappeared after WO, and again was significantly reduced after T3 (Fig. 1c and see Supplementary Fig. 2d). At T2, J20 mice showed learning acquisition deficits in the Barnes maze spatial memory training phase evidenced by increased primary errors to identify the target correctly compared to WT mice. This deficit was not significantly attenuated by VX-765 (Fig. 1d). Primary latency to find the target did not differ between the three groups (see Supplementary Fig. 2e). During the probe test, primary latency, primary errors (Fig. 1e) and the ability to find the target (Fig. 1f) were clearly impaired in vehicle-injected J20 compared to WT mice. VX-765 eliminated these spatial memory deficits in J20 mice. After WO, vehicle-injected J20 showed learning deficits early on, but all groups performed equally well by the end of training (Fig. 1g and see Supplementary Fig. 2f). The difference in WO primary latency compared to T2 is likely due to a learning effect by repetition or a reduction of stress to the apparatus. VX-765-injected, but not vehicle-injected, J20 mice were normal in primary latency and errors during the probe, suggesting spatial memory retention even after 1 month without drug (Fig. 1h). VX-765-injected mice also performed better than vehicle-injected J20 mice in their ability to find the target (Fig. 1i). J20 mice performance in the Y maze working memory task was consistently low but not always statistically different from WT or treated J20 mice (see Supplementary Fig. 2g).

VX-765 was administered at 25 and 10 mg kg$^{-1}$ to assess the dose-response of J20 to VX-765 (Fig. 2). All groups showed normal motivation (see Supplementary Fig. 3a) and thigmotaxic behaviour (see Supplementary Fig. 3b). NOR discrimination index was normalized with 25 mg kg$^{-1}$ dose at T1 and T2, deficits reappeared after WO, and returned to normal at T3, similar to the results with the 50 mg kg$^{-1}$ dose (Fig. 2a and see Supplementary Fig. 3c). Mice treated with 10 mg kg$^{-1}$ showed normalized NOR behaviour only after T2, the effect was lost with WO, and reappeared after T3. J20 hyperactivity showed a nonsignificant improvement at T2 with 25, but not 10, mg kg$^{-1}$ (see Supplementary Fig 3d, e). In the T2 Barnes maze, learning acquisition did not change after VX-765 treatment by the end of the training phase (Fig. 2b and see Supplementary Fig. 3f), and J20 remained impaired in the probe primary latency at both the 25 and 10 mg kg$^{-1}$ dose (Fig. 2c). However, primary errors were

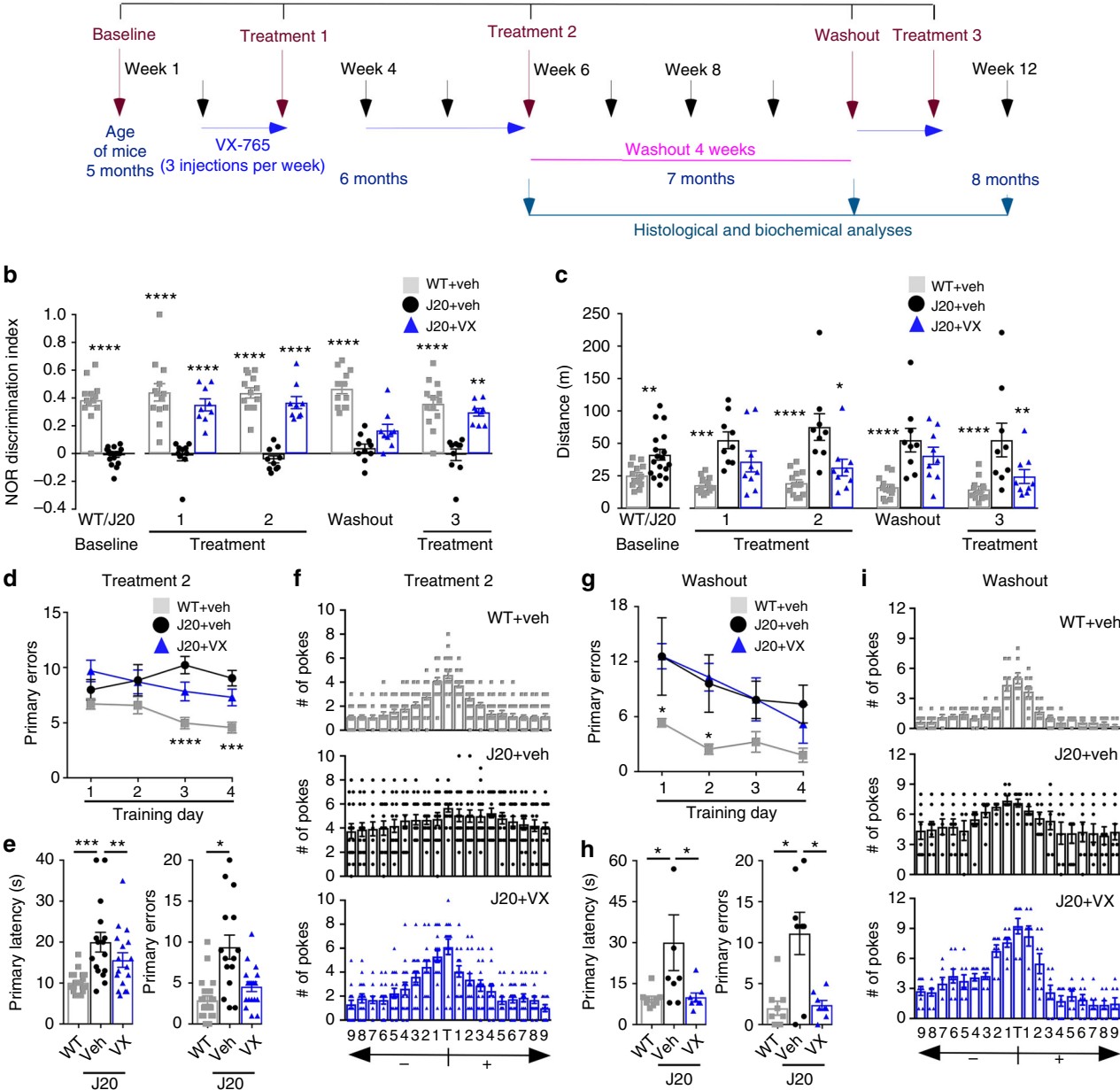

**Fig. 1** VX-765 treatment restores J20 mice cognitive function. **a** Experimental treatment paradigm. **b** NOR Discrimination index from vehicle-treated WT (grey squares), vehicle-treated J20 (black circles) and VX-765-treated J20 (blue triangles). Each mouse tested is represented by one symbol. Data represent mean and s.e.m. (Treatment, $F_{(2,20)} = 85.8$, $p < 0.0001$; Time, $F_{(4,80)} = 4.188$, $p = 0.0039$; Treatment × Time, $F_{(8,80)} = 3.599$, $p = 0013$). **c** Distance travelled during open field task (Treatment, $F_{(2,19)} = 11.47$, $p = 0.0005$; Treatment × Time, $F_{(8,76)} = 5.69$, $p < 0.0001$). **b**, **c** Two-way repeated-measures ANOVA and Dunnett's post-hoc versus J20 + vehicle, *$p < 0.05$, **$p < 0.01$, ***$p < 0.001$, ****$p < 0.0001$. **d–i** Barnes maze: learning acquisition # of primary errors during **d** T2 (Treatment, $F_{(2,212)} = 19.93$, $p < 0.0001$) and **g** WO (Treatment, $F_{(2,56)} = 10.86$, $p = 0.001$; Training day, $F_{(3,56)} = 3.392$, $p < 0.0241$). **d**, **g** Two-way repeated-measures ANOVA and Dunnett's post-hoc versus J20 + vehicle *$p < 0.05$, ***$p < 0.001$, ****$p < 0.0001$. Probe primary latency and errors during **e** T2 and **h** WO. Target preference: # of pokes of each hole labelled +1 to +9 to the right or −1 to −9 to the left of the target (T) during the probe after **f** T2 and **i** WO (T2 primary latency, $F_{(2,52)} = 5.879$, $p = 0.0050$; T2 primary errors, $F_{(2,52)} = 9.998$, $p = 0.0002$; WO primary latency, $F_{(2,22)} = 4.076$, $p = 0.0312$; WO primary errors, $F_{(2,22)} = 10.84$, $p = 0.0005$). **e**, **h** ANOVA, Tukey's post-hoc, *$p < 0.05$, **$p < 0.01$, ***$p < 0.001$

decreased with 25 mg kg$^{-1}$ and both doses showed that the J20 mice had improved ability to recognize the target during the probe (Fig. 2d). After WO, primary latency was shorter initially during training (see Supplementary Fig. 3g), but no differences in learning were observed among the three groups by the end of the training (Fig. 2e). No differences were observed in primary latency and errors, with or without drug during the probe

(Fig. 2f). However, mice treated with 25 mg kg$^{-1}$, but not 10 mg kg$^{-1}$, retained their ability to recognize the target (Fig. 2g). Together, these results demonstrate a VX-765 dose-dependent effect in reversing J20 episodic and spatial memory deficits.

To confirm that VX-765's effect on re-establishing normal cognition in J20 mice was due to Casp1 inhibition, J20 mice were generated with a *Casp1* null background (J20/*Casp1*$^{-/-}$) and

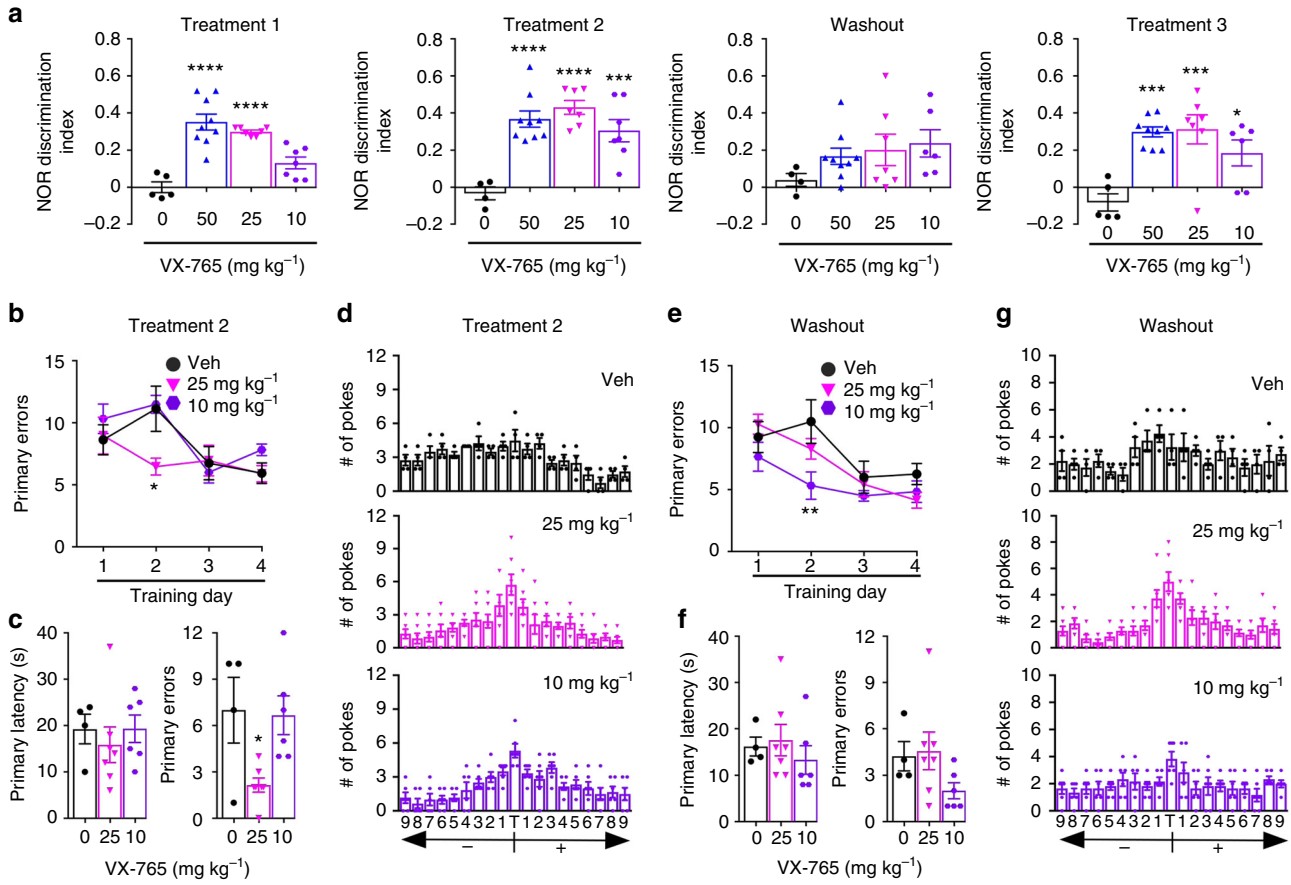

**Fig. 2** VX-765 dose-dependently restores cognitive function in J20 mice. **a** NOR discrimination index at T1 (F(3,24) = 20.42, $p < 0.0001$), T2 (F(3,23) = 12.83, $p < 0.0001$), WO, and T3 (F(3,23) = 8.828, $p = 0.0004$). Vehicle-treated (black circles), and 50 mg kg$^{-1}$ (blue triangles), 25 mg kg$^{-1}$ (pink triangles), or 10 mg kg$^{-1}$ (purple hexagon) VX-765-treated J20 mice. Each mouse tested is represented by one symbol. Data represent mean and s.e.m. **b**–**g** Barnes maze during **b**–**d** T2 and **e**–**g** WO. Learning acquisition at **b** T2 and **e** WO; probe and target preference at **c**, **d** T2 and **f**, **g** WO. Learning acquisition # of primary errors during **b** T2 (Treatment, F(2,56) = 3.377, $p = 0.0412$; Training day, F(3,56) = 7.102, $p = 0.0004$), and **e** WO (Treatment, F(2,56) = 5.434, $p = 0.007$; Training day, F(3,56) = 11.52, $p < 0.0001$), probe primary errors at **c** T2 (F(2,14) = 5.69, $p = 0.0155$). **a**, **c**, **f** ANOVA and **b**, **e** Two-way repeated-measures ANOVA, Dunnett's post-hoc versus J20 + vehicle, *$p < 0.05$, **$p < 0.01$, ***$p < 0.001$, ****$p < 0.0001$

behaviourally assessed at 8 months of age, corresponding to the age of J20 mice after VX-765 T3 (Fig. 3). Similar to 5-month-old J20 mice, 8-month-old J20/Casp1$^{-/-}$ showed normal locomotor activity and thigmotaxis (see Supplementary Fig. 4a, b) compared to J20/WT, J20/Casp1$^{+/-}$, WT/WT, WT/Casp1$^{-/-}$, or WT/Casp1$^{+/-}$. NOR discrimination index, which was strongly decreased in J20//WT mice, was normalized in J20/Casp1$^{-/-}$ and J20/Casp1$^{+/-}$ (Fig. 3a and see Supplementary Fig. 4c). Mice hyperactivity was similar in J20/WT, J20/Casp1$^{-/-}$, and J20/Casp1$^{+/-}$ mice, in contrast to observations in the VX-765 studies (Fig. 3b). This indicates that the J20 hyperactivity is exacerbated by the extra manipulations or injections in the VX-765 study. Partial or complete Casp1 knockout (KO) ameliorated J20 mice learning deficits at days 3 and 4 of the Barnes maze learning acquisition phase (Fig. 3c), but there were no differences in primary latency to find the target between groups (see Supplementary Fig. 4d). During the probe, primary latency did not differ between the different genotypes (Fig. 3d); however, primary errors and the ability to find the target was impaired in J20/WT mice and normalized after partial or complete Casp1 KO (Fig. 3d, e). Together, these results suggest that VX-765 reverses episodic and spatial memory deficits by inhibiting Casp1 in J20 mice.

**VX-765 reverses neuroinflammation in J20 mice brains.** Iba1 is a marker of activated microglia and inflammation[33]. Increased

Iba1-positive microglia in the hippocampus and cortex of 8-month-old vehicle-treated J20 mice, compared to vehicle-treated WT mice, were normalized in VX-765-treated J20 mice, indicating that VX-765 reversed inflammation (Fig. 4a). This was confirmed quantitatively as the number of Iba1-immunopositive microglia measured from the pyramidal cell layer to the stratum lacunosum moleculare (SLM) layer of the hippocampal CA1 region, or from the cortical retrosplenial area to the S1, returned to normal after VX-765 treatment (Fig. 4b). Quantitation of microglial activation levels based on morphological measurements[34] (see Supplementary Fig. 5a) showed decreased type I resting microglia and increased type II and III activated microglia in vehicle-treated J20 hippocampus and cortex compared to vehicle-treated WT and VX-765-treated J20 mice (Fig. 4b). Hippocampal Il-1β tended towards an increase in vehicle-treated J20 compared to VX-765-treated J20, vehicle-treated WT, and 5-month-old J20 baseline hippocampus (Fig. 4b). Cortical Il-1β tended towards an increase in vehicle-treated WT and J20 compared to baseline or VX-765-treated J20. No significant difference was observed in plasma Il-1β level in VX-765-treated mice (see Supplementary Fig. 5b) or other inflammatory brain protein levels, although TNF-α, KC-GRO, and IFN-γ were slightly elevated in vehicle-treated J20 and normalized after VX-765 treatment (see Supplementary Fig. 5c).

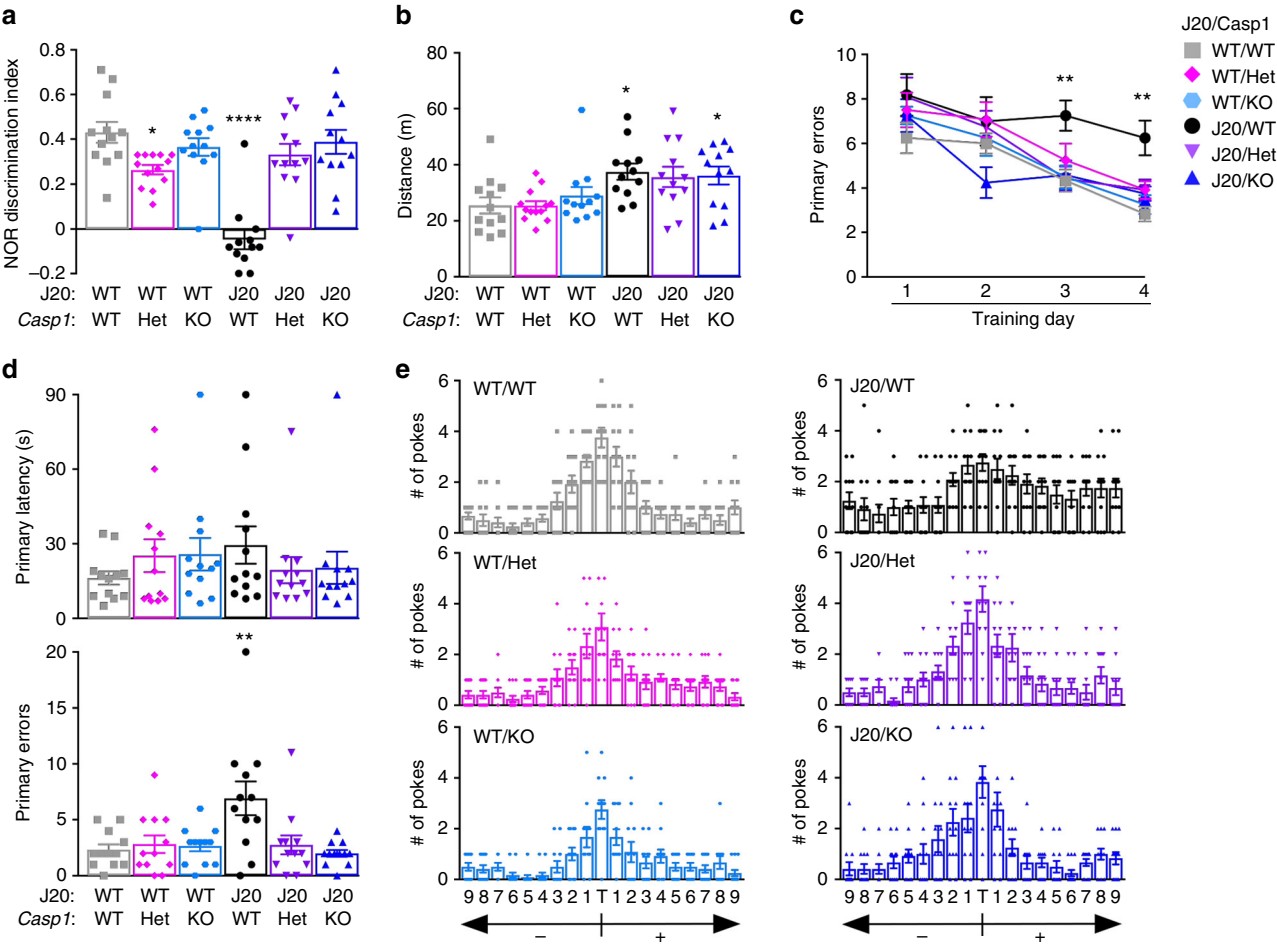

**Fig. 3** *Casp1* KO restores cognitive function in J20 mice. **a–e** Genotypes: J20$^{-/-}$/*Casp1*$^{-/-}$ WT/WT (grey squares), J20$^{-/-}$/*Casp1*$^{+/-}$ WT/Het (pink diamonds), J20$^{-/-}$/*Casp1*$^{-/-}$ WT/KO (blue symbol), J20$^{-/+}$/*Casp1*$^{+/+}$ J20/WT (black circles), J20$^{-/+}$/*Casp1*$^{-/+}$ J20/Het (purple triangles), J20$^{-/+}$/*Casp1*$^{-/-}$ J20/KO (blue triangles). Each mouse tested is represented by one symbol. Data represent mean and s.e.m. **a** NOR discrimination index (F (5,67) = 16.22, p < 0.0001) and **b** distance travelled during open field task (F(5,67) = 3.717, p = 0.005). **c–e** Barnes maze: **c** # of primary errors during learning acquisition (Genotype, F(5,263) = 6.469, p < 0.0001; Training day, F(3,263) = 29.44, p < 0.0001), **d** probe primary latency and errors (Primary errors, F(5,66) = 4.8, p = 0008) and **e** target preference. **a**, **b**, **d** ANOVA and **c** two-way ANOVA, Dunnett's post-hoc versus WT/WT. *p < 0.05, **p < 0.01, ****p < 0.0001

Analyses of activated microglia were conducted in the J20/WT, WT/*Casp1*$^{-/-}$ and J20/*Casp1*$^{-/-}$ to confirm that VX-765 effects occurred through inhibition of Casp1. The microglia in J20/*Casp1*$^{-/-}$ were as activated as those of J20/WT (Fig. 4a and see Supplementary Fig. 5d) but as observed in VX-765-treated J20 (Fig. 4b), the number of Iba1-positive microglia increased in J20/WT compared to WT and were normalized in J20/*Casp1*$^{-/-}$ (Fig. 4c). However, in contrast to VX-765 treatments, the absence of *Casp1* did not modify the distribution of the subtypes of microglia; fewer resting microglia at the expense of increased activated microglia was observed in J20/*Casp1*$^{-/-}$ mimicking levels of the J20/WT. Il-1β levels were unchanged in J20/*Casp1*$^{-/-}$ compared to J20/WT and surprisingly these were not different from WT mice. These results suggest that cremophor vehicle injections in the VX-765 study may have an inflammatory effect that exacerbates APP-mediated inflammation in J20.

Increased GFAP immunopositive astrogliosis in the hippocampus and cortex of J20 mice was normalized to WT levels with VX-765 treatment and absence of *Casp1* (Fig. 4d–f). Overall, these results indicate that VX-765 treatment reversed both microglial and astroglial activation in J20 brains.

**VX-765 prevents Aβ accumulation in J20 mice brains**. To determine if APP or Aβ levels were altered by VX-765 treatment or the absence of *Casp1* in J20 mice, western blots against full length and C-terminal fragments (CTF) of APP, immunohistochemistry with an anti-Aβ$_{1-40}$ antiserum, and ELISA measurement of Aβ$_{38, 40, and 42}$ were conducted. Thioflavin S staining of histological brain sections confirmed the amyloid status of Aβ immunopositivity. Thioflavin S and immunopositive Aβ levels were substantially reduced in VX-765-treated compared to vehicle-injected J20 hippocampus and cortex (Fig. 5a, b and see Supplementary Fig. 6a). However, Aβ deposits did not completely disappear and were comparable to those observed in baseline J20 brains (Fig. 5a). These remaining Aβ deposits were mainly localized in the SLM and dentate gyrus regions of the hippocampus (see Supplementary Fig. 6a). Levels of immunopositive Aβ plaques were similarly reduced in the J20/*Casp1*$^{-/-}$ hippocampus and cortex (Fig. 5a, c and see Supplementary Fig. 6b). In addition, a much lower level of Aβ was observed in untreated J20 (J20/WT; Fig. 5c) compared to vehicle-injected J20 (Fig. 5b), suggesting an exacerbation of Aβ deposition with the cremophor vehicle injections.

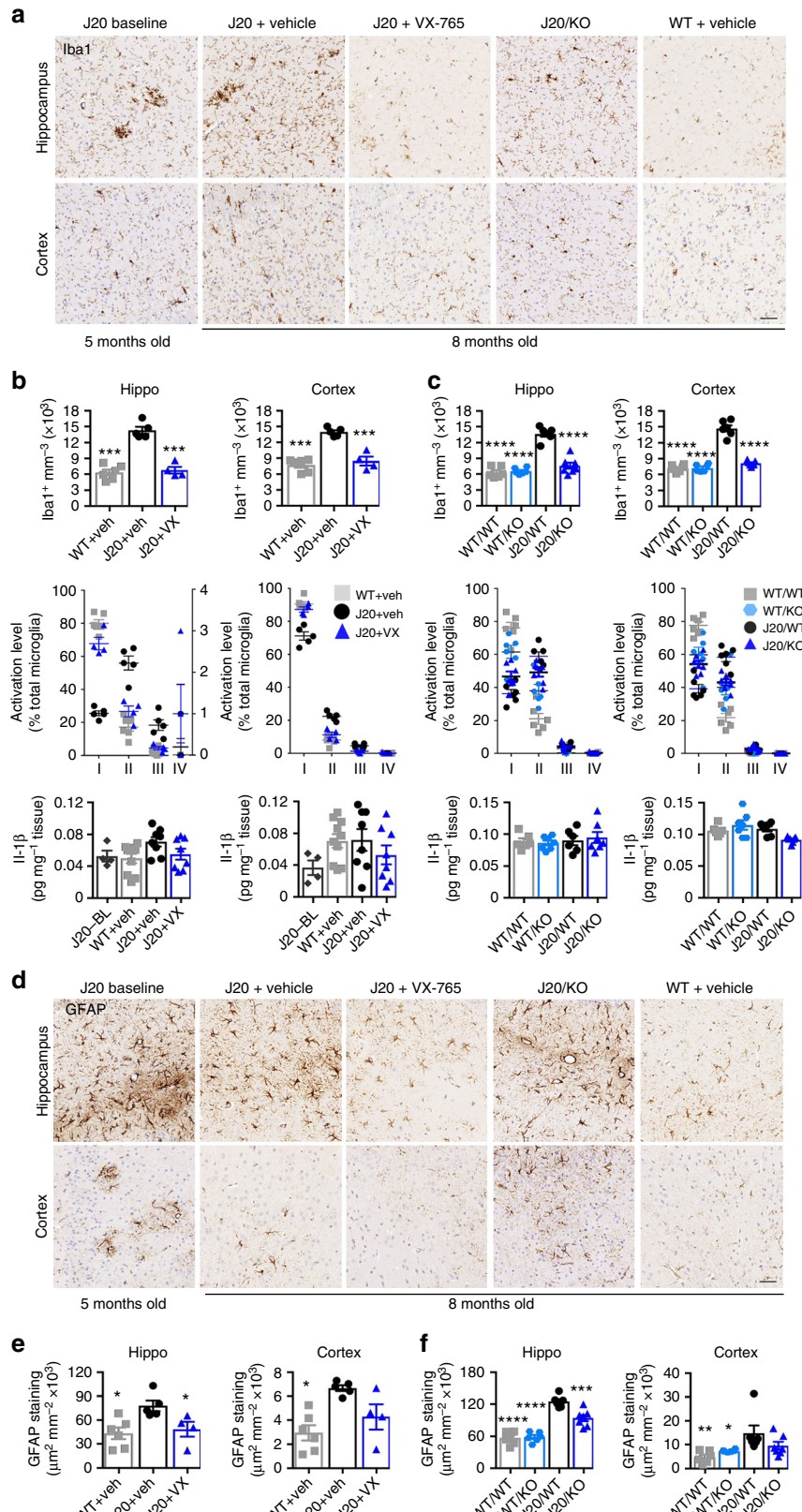

RIPA-soluble $A\beta_{42}$/total $A\beta_{38+40+42}$ ratios were reduced to baseline J20 levels in the hippocampus and less obviously in the cortex of VX-765-treated mice (Fig. 5d), whereas total $A\beta$ levels did not change (Fig. 5e). Formic acid-soluble $A\beta_{42}$/total $A\beta_{38+40+42}$ ratios were reduced in the hippocampus, but not in the cortex (Fig. 5f), whereas total $A\beta$ levels were more variable than RIPA

$A\beta$ levels but tended towards a reduction in both hippocampus and cortex (Fig. 5g). VX-765 treatment had no significant effect on $A\beta_{38}$ or $A\beta_{40}$ subtypes in hippocampus and cortex (see Supplementary Fig. 6c, d). The reduction in $A\beta_{42}$ levels with VX-765 treatment was not due to decreased APP mRNA (see Supplementary Fig. 7a) or protein levels measured with the

**Fig. 4** VX-765 reverses neuroinflammation in J20 mice. **a** Iba1-immunopositive microglia in hippocampal SLM and cortical S1 regions. **b, c, e** Vehicle-treated J20$^{-/-}$/Casp1$^{-/-}$ WT (grey squares), vehicle-treated J20$^{-/-}$/Casp1$^{-/-}$ WT/KO (blue hexagons), J20$^{-/+}$ at 5 months baseline (J20-BL; dark grey hexagons), vehicle-treated J20$^{-/+}$ (black circles), and VX-765-treated J20$^{-/+}$ (purple triangles). **b** Stereological quantification of Iba1-positive microglia from hippocampal pyramidal cell layer to the SLM (F(2,11) = 58, $p < 0.0001$) and cortex (F(2,11) = 39.95, $p < 0.0001$) (top panels), average % distribution of morphological microglial subtypes I, II, III and IV (middle panels), and Il-1-β levels (bottom panels) in 5-month-old baseline J20 and in 8-month-old WT + vehicle, J20 + vehicle, and J20 + VX-765 WT mice. ANOVA, Dunnett's post-hoc versus J20 + vehicle, ***$p < 0.001$. **c** Stereological quantification of Iba1-positive microglia in the hippocampal CA1 (F(3,21) = 50.53, $p < 0.0001$) and cortex (F(3,21) = 96.21, $p < 0.0001$) (top panels), average % distribution of morphological microglial subtypes I, II, III and IV (middle panels), and Il-1-β levels (bottom panels) in WT/WT, WT/Casp1$^{-/-}$ (WT/KO), J20/WT, and J20/Casp1$^{-/-}$ (J20/KO) mice. ANOVA, Dunnett's post-hoc versus J20/WT, ****$p < 0.0001$. **d** Micrographs of GFAP immunopositive astrocytes. **e** GFAP immunostaining density in vehicle-treated WT and J20 and in VX-765-treated J20 brain hippocampi (F(2,12) = 5.234, $p = 0.0232$) and cortex (F(2,12) = 8.582, $p = 0.0049$). **f** GFAP immunostaining density in J20/KO and littermate control brain hippocampi (F(3,21) = 41.15, $p < 0.0001$) and cortex (F(3,21) = 4.746, $p = 0.0111$). ANOVA, Dunnett's post-hoc versus J20 + vehicle (**e**) or J20/WT (**f**), *$p < 0.05$, ***$p < 0.001$, ****$p < 0.0001$. Scale bar in **a, d** = 50 μm

human-specific 6E10 antibody (Fig. 5l, m) or the mouse and human anti-CTF antibody (see Supplementary Fig. 7c, d).

RIPA-soluble Aβ$_{42}$/total Aβ$_{38+40+42}$ ratios were reduced in the hippocampus and cortex in J20/Casp1$^{+/-}$ and J20/Casp1$^{-/-}$ mice, with no changes in total Aβ levels although there was a trend towards an increase in the hippocampus (Fig. 5h, i). These reduced ratios were due to increased Aβ$_{38}$ and Aβ$_{40}$ rather than decreased Aβ$_{42}$ levels in J20/Casp1$^{-/-}$ (see Supplementary Fig. 6e). J20/Casp1$^{+/-}$ and J20/Casp1$^{-/-}$ mice had similar formic acid-soluble Aβ$_{42}$/total Aβ$_{38+40+42}$ ratios, total Aβ levels (Fig. 5j, k), and Aβ subtypes in the hippocampus or cortex (see Supplementary Fig. 6f). The absence of Casp1 in J20 mice had no effect on APP protein levels (Fig. 5n, o and see Supplementary Fig. 7e, f).

To determine if Aβ clearance might have increased with VX-765 treatment or the absence of Casp1, we measured the levels of Aβ-degrading neprilysin (Nep) and insulin degrading (IDE) enzymes. Increased degradation of Aβ was unlikely since IDE or Nep protein (see Supplementary Fig. 8a, b) and mRNA levels (see Supplementary Fig. 8c, d) were unchanged with VX-765 treatment or the absence of Casp1 (see Supplementary Fig. 8e, f). Together, these results indicate that VX-765 treatment and the absence of Casp1 stopped the progressive deposition of Aβ in J20 brains.

**VX-765 does not alter Casp8 or Casp9 processing in J20 mice.** Casp5, Casp8, Casp9 and Casp10, which were inhibited in the nanomolar range in vitro by VX-765 and VRT-043198, were considered to determine whether VX-765 could affect other caspases in J20 mice (see Supplementary Fig. 9). Since Casp5 and Casp10 are not expressed in mouse[35], they were not examined. Pro-Casp9 levels were significantly reduced in the hippocampus, but not in cortex, of vehicle-treated J20 compared to WT mice, but normalized after VX-765 treatment (see Supplementary Fig. 9a). This could indicate that pro-Casp9 has been converted to active subunits, and that activation was inhibited by VX-765. Yet, activated Casp9 subunits could not be detected in the mouse brain. Since Casp3 is an excellent substrate of Casp9, we verified if Casp3 was processed into its active subunits in the hippocampal tissues. No active Casp3 subunit was observed in vehicle- or VX-765-treated J20 hippocampus, and there was also no effect of VX-765 on pro-Casp3 levels. Cleaved Casp8 20 kDa active subunit could not be detected in the hippocampus in either WT or J20 mice. A faint band corresponding to 43 kDa cleaved Casp8 active subunit (p43) was detected in the cortex, but its levels remained unaffected after VX-765 treatment (see Supplementary Fig. 9a).

Because processed Casp8 and Casp9 levels in the adult mouse brain were low, VX-765's potential effects on Casp8 and Casp9 were further analysed in the spleen, which expresses higher levels

of these enzymes and is known to naturally contain abundant active Casp8 (see Supplementary Fig. 9b, c). There were no differences in active Casp8 subunits, pro-Casp9 levels, or Casp8 or Casp9 substrate pro-Casp3 and active Casp3 20 kDa subunit, among the three groups (see Supplementary Fig. 9c). Together, these results suggest that VX-765 did not inhibit Casp8 and Casp9 in J20 mice.

**VX-765 normalizes synaptophysin in J20 mice brains.** At 5 months of age, J20 hippocampi showed decreased immunopositive synaptophysin levels (Fig. 6a). Synaptophysin levels remained low in 8-month-old vehicle-treated J20 hippocampus but were significantly increased and returned to normal levels in VX-765-treated mice hippocampi (Fig. 6a, b). There was no significant difference in the cortex. Measures of 84 different synaptic gene mRNA levels in vehicle-treated J20 hippocampi revealed significantly increased mRNA levels in three genes involved in synaptic function: Camk2a, Grin2b, Kif17 (Fig. 6c). Levels were decreased with VX-765 treatment. Additionally, TNF-α mRNA was increased in J20 and VX-765-treated J20 hippocampi (Fig. 6c), whereas TNFα protein levels increased slightly in J20 mice hippocampi and returned to normal with the VX-765 treatment (see Supplementary Fig. 5c). These results indicate a normalizing effect of VX-765 on several synaptic components that may account for the reversal to normal cognition.

**VX-765 prevents axonal degeneration in human neurons.** To determine if VX-765 can protect neurons against neurodegeneration, VX-765 was assessed in human CNS human primary neuron cultures (HPN). Treatment of HPN with 25, 50, 100, or 200 μM VX-765 for 72 h was not toxic (Fig. 7a). As previously described[19], EGFP was homogeneously distributed within the cell body and neurites of HPN, whereas coexpression with APP$^{WT}$ resulted in EGFP-positive beading indicative of neurodegeneration (Fig. 7b). HPN were pretreated with VX-765 for 1 h before APP transfection and treatment continued thereafter. VX-765 treatment at 25 and 50 μM prevented neuritic beading in APP$^{WT}$-transfected neurons at 48 and 72 h post transfection, similar to the Casp1 Z-YVAD-fmk peptide inhibitor (Fig. 7c). Similarly, VX-765 protected against, albeit less strongly, serum-deprivation-induced neuritic beading. To assess if neuritic beading was reversible, VX-765 was administered 48 h after APP$^{WT}$ transfection or serum deprivation. VX-765 treatment did not reverse but prevented further neuritic beading in APP-transfected neurons. In contrast, Z-YVAD-fmk was unable to reverse or prevent neuritic beading at any time point. The 50 μM VX-765 reduced secreted and cellular Aβ$_{42}$/Aβ$_{38+40+42}$ levels, whereas 25 μM reduced secreted, but not cellular, Aβ$_{42}$/total Aβ levels

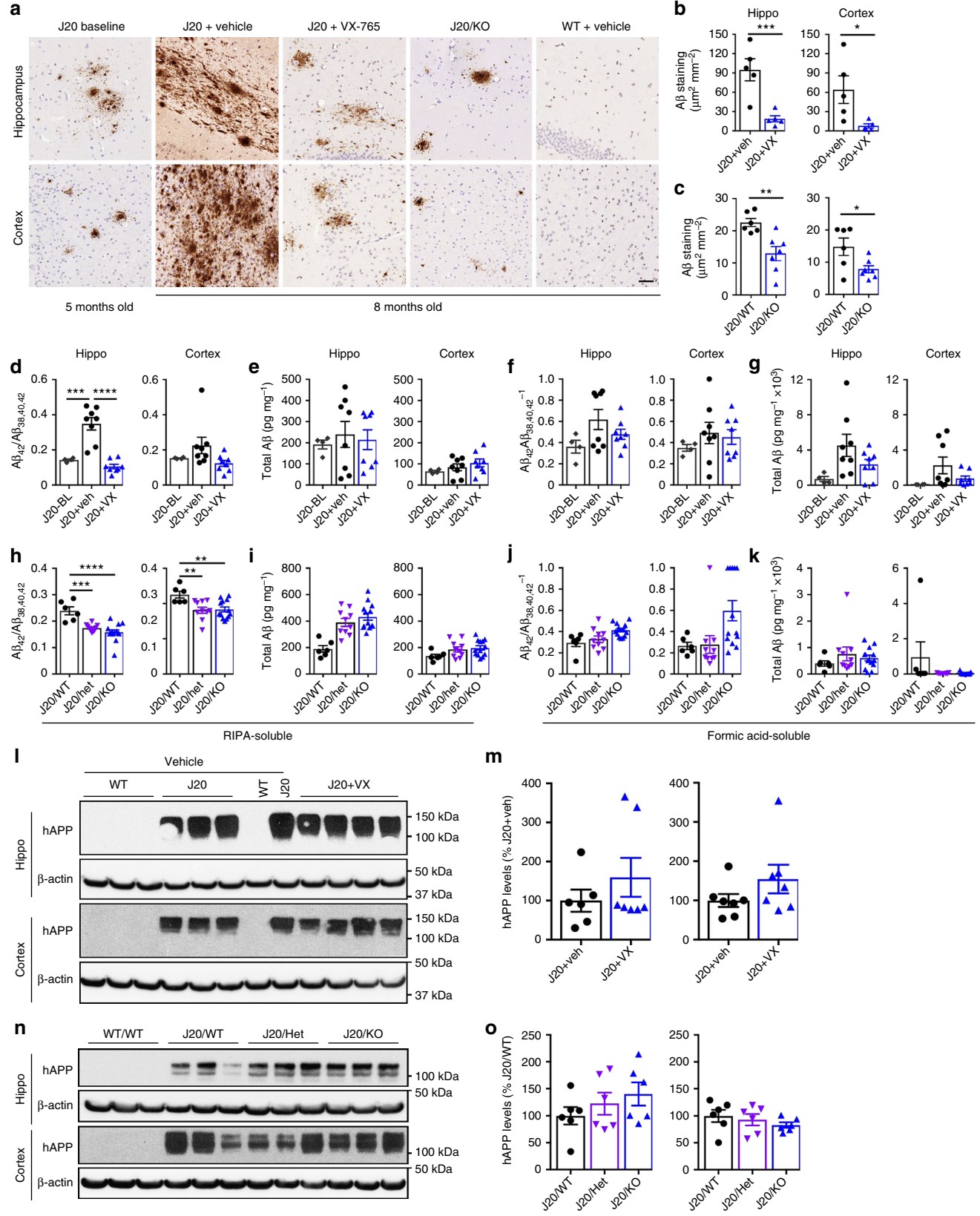

(Fig. 7d). Il-1β and TNF-α levels were significantly increased after serum deprivation but reduced after VX-765 (Fig. 7e). These results indicate that VX-765 can prevent human neuronal degeneration.

## Discussion

These results strongly indicate that VX-765 is a promising drug against AD. Our idea of using Casp1 as a therapeutic target is based on the identification of the Nlrp1-Casp1-Casp6 neuronal

**Fig. 5** VX-765 prevents progressive Aβ accumulation in J20 mice. **a** Aβ micrographs of hippocampal SLM and S1 cortex (scale bar = 50 μm). **b** Quantitative analysis comparing Aβ immunostaining density between J20 + vehicle (black circles) and J20 + VX-765 (blue triangles) mice in the CA1 hippocampal pyramidal cell layer to the SLM ($p = 0.0029$) and cortex ($p = 0.0314$, unpaired $t$ test). **c** Quantitative analysis comparing Aβ immunostaining density between $J20^{-/+}/Casp1^{+/+}$ J20/WT (black circles) and $J20^{-/+}/Casp1^{-/-}$ J20/KO (blue triangles) mice in the CA1 hippocampal cell layer ($p = 0.0040$) and cortex (0.0276, unpaired $t$ test). **d−g** RIPA- (**d**, **e**) and formic acid- (**f**, **g**) soluble Aβ levels in the hippocampus and cortex of 5-month baseline $J20^{-/+}$ (black diamonds), J20 + vehicle (black circles), and J20 + VX-765 (blue triangles) mice. **d**, **f** $Aβ_{42}/Aβ_{38} + Aβ_{40} + Aβ_{42}$ levels (RIPA soluble, hippo: $F_{(2,17)} = 27.85$, $p < 0.0001$) and **e**, **g** total Aβ ANOVA, Dunnett's post-hoc versus J20 + vehicle, ***$p < 0.001$, ****$p < 0.0001$. **h−k** $J20^{-/+}/Casp1^{+/+}$ J20/WT (black circles), $J20^{-/+}/Casp1^{-/+}$ J20/Het (purple triangles), and $J20^{-/+}/Casp1^{-/-}$ J20/KO (blue triangles) RIPA- (**h**, **i**) and formic acid- (**j**, **k**) soluble Aβ levels in the hippocampus and cortex of J20/WT, J20/$Casp1^{+/-}$ (J20/Het), and J20/$Casp1^{-/-}$ J20/KO mice. **h**, **j** $Aβ_{42}/Aβ_{38} + Aβ_{40} + Aβ_{42}$ levels (RIPA-soluble, hippo: $F_{(2,26)} = 16.03$, $p < 0.0001$; RIPA-soluble, cortex: $F_{(2,26)} = 6.602$, $p < 0.0048$) and **i**, **k** total Aβ ANOVA, Dunnett's post-hoc versus J20/WT, **$p < 0.01$, ***$p < 0.001$, ****$p < 0.0001$. **l−o** Human APP protein levels (6E10 immunostaining) and quantification in the hippocampus and cortex of **l**, **m** J20 + vehicle (black circles) and J20 + VX-765 mice (blue triangles), and **n−o** J20/WT (black circles), J20/Het (purple triangles), and J20/KO (blue triangles) mice

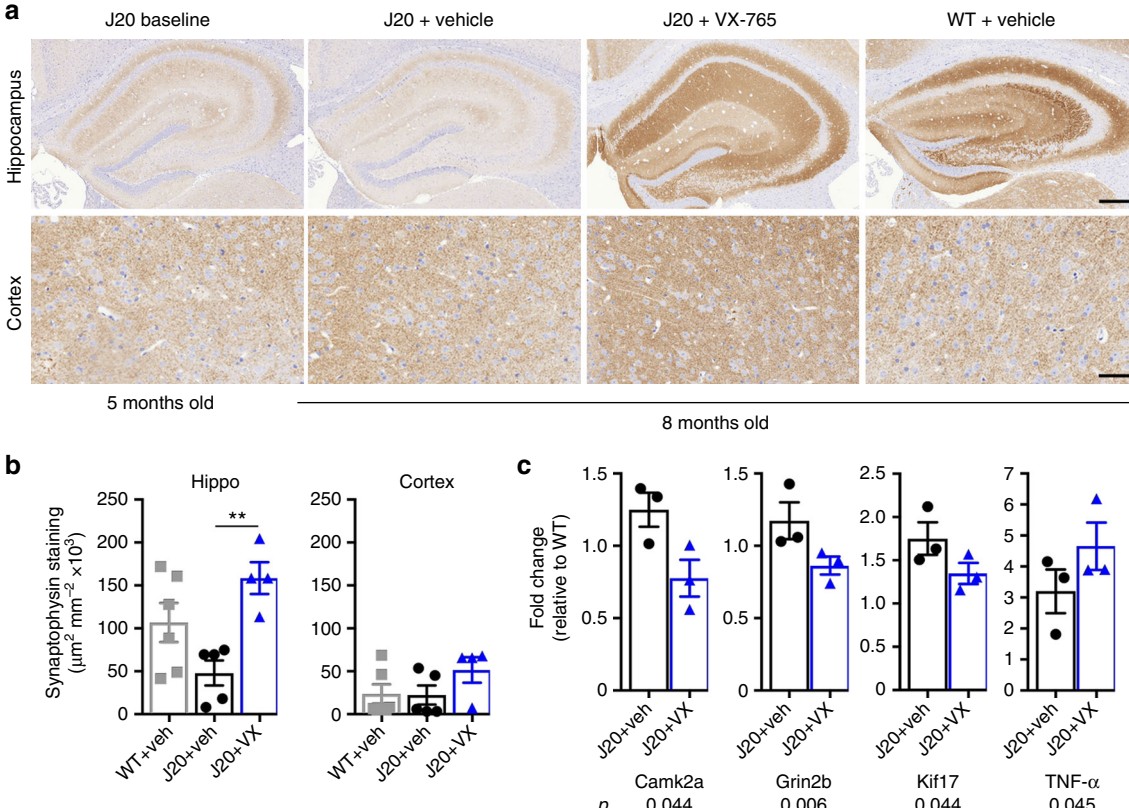

**Fig. 6** VX-765 reverses loss of synaptophysin in J20 mice. **a** Synaptophysin immunopositive micrographs. Scale bar = 200 μm for hippocampus and 50 μm for cortex. **b** Synaptophysin immunopositive density in WT + vehicle (grey squares, $n = 4$), J20 + vehicle (black circles, $n = 4$), and J20 + VX-765 (blue triangles, $n = 4$) hippocampus ($F_{(2,9)} = 7.974$, $p = 0.0102$, ANOVA, Tukey's post-hoc, **$p < 0.01$) and cortex. **c** Significantly altered (Kruskall−Wallis) synaptic protein mRNA levels in vehicle or VX-765-treated J20 mice hippocampi ($n = 3$ per group)

degeneration pathway in CNS primary human neurons, the association of this pathway with memory and cognitive impairment in aged humans and mice[11–16,36,37], and the genetic association between *NLRP1* and *CASP1* with AD progression or AD[38,39]. VX-765 effect is exquisitely rapid against episodic and spatial memory impairment, which are normalized after only 1 (3 injections) or 3 (9 injections) weeks of treatment, respectively. Reversal of cognitive impairment is accompanied by normalized microglial and astroglial reactivity and synaptophysin immunohistochemical staining, a normalization of gene expression of four different synaptic components, and the prevention of progressive amyloid pathology in the brain. Target engagement for VX-765 against Casp1 was confirmed in the J20 *Casp1* null mice, which also exhibited normal episodic and spatial memory, and reduced

inflammation and Aβ accumulation. To our knowledge, this drug provides the most rapid response observed with a preclinical treatment in the J20 AD mouse model[40–42].

Casp1 represents a feasible therapeutic target against age-dependent cognitive impairment and AD. Rare naturally occurring human *CASP1* variants associated with very low activity, in vitro and in vivo, suggest that inhibition of Casp1 activity is compatible with normal human life[44,45]. In these *CASP1* variants, proinflammatory signalling still occurs through NFKβ[46]. *Casp1* null mice are fertile and healthy indicating that Casp1 is not essential for proper development[47]. The implication of microglia in AD progression has been nicely demonstrated in the *APP/PS1* AD transgenic mouse model where Aβ-mediated activation of microglial Nlrp3 inflammasome results in cognitive deficits and

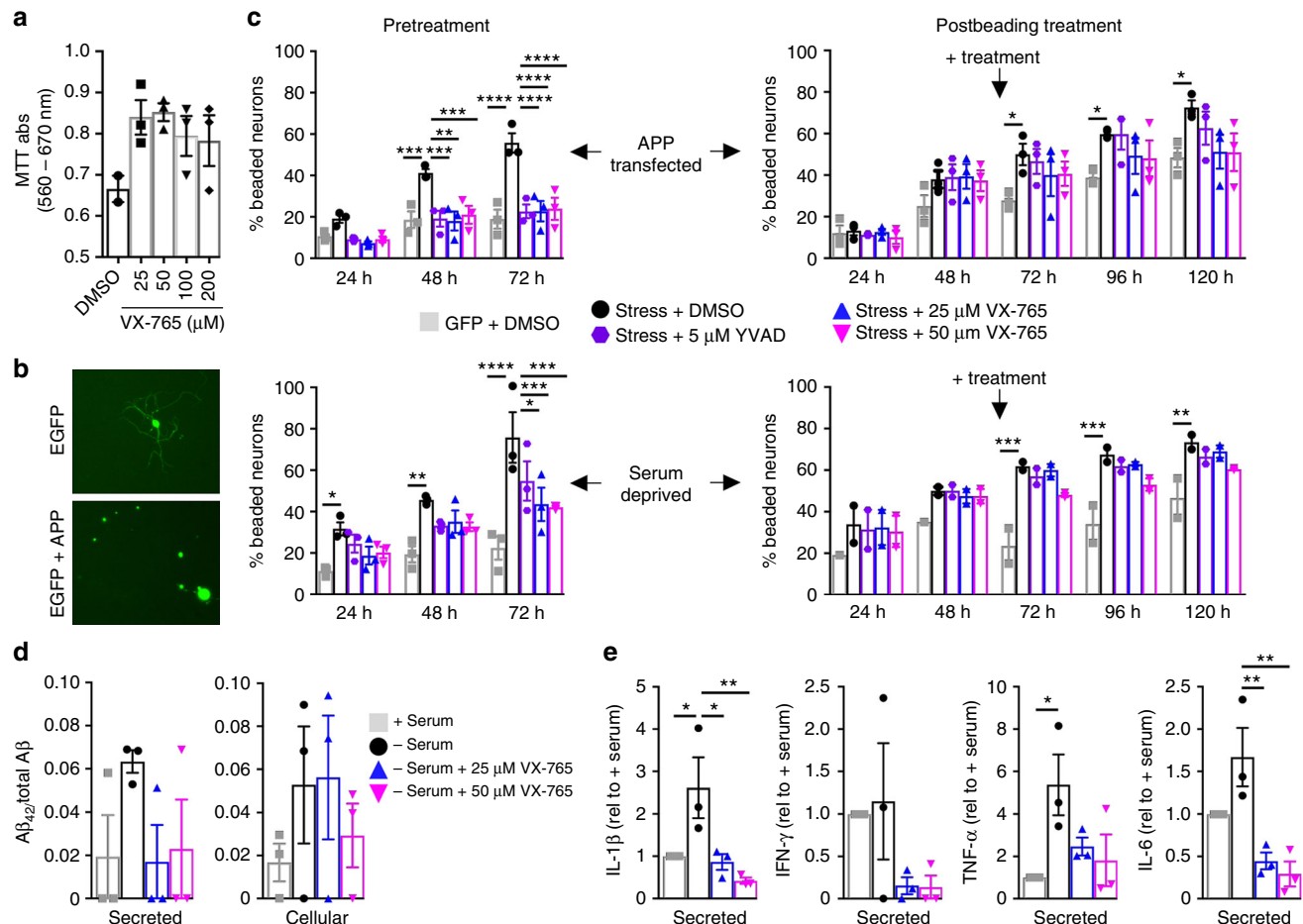

**Fig. 7** VX-765 protects human neurons against stress-mediated neuritic beading. **a** MTT assay. **b** EGFP fluorescent micrographs of EGFP- or APP+ EGFP-transfected neurons. **c** % beading in 25 or 50 μM VX-765 or 5 μM Z-YVAD-fmk Casp1 peptide inhibitor pretreated (1 h) or post-treated (48 h) APP-transfected or serum-deprived neurons. DMSO vehicle treatment of EGFP-transfected neurons (grey squares), DMSO- (black circles), 5 μM YVAD- (purple hexagons), 25 μM VX-765- (blue triangles) or 50 μM VX-765- (purple triangles)-treated stressed (APP transfection in upper panels or serum deprived in bottom panel) neurons. Two-way repeated-measures ANOVA, Dunnett's post hoc versus $APP^{WT}$ or serum-deprived, $*p < 0.05$, $**p < 0.01$, $***p < 0.001$, $****p < 0.0001$. Pretreatment: $APP^{WT}$ (Treatment $F_{(4,10)} = 10.17$, $p = 0.0015$; Time $F_{(2,20)} = 93.32$, $p < 0.0001$; Treatment × Time $F_{(8,20)} = 6.736$, $p = 0.0003$), serum-deprived (Treatment $F_{(4,10)} = 10.2$, $p = 0.0015$; Time $F_{(2,20)} = 37.65$, $p < 0.0001$). Postbeading treatment: $APP^{WT}$ (Time $F_{(4,40)} = 102.7$, $p < 0.0001$), serum-deprived (Treatment $F_{(4,23)} = 14.12$, $p < 0.0001$; Time $F_{(4,23)} = 24.35$, $p < 0.0001$). **d, e** Serum-treated normal (grey square), serum-deprived (black circles), serum-deprived and treated with 25 μM VX-765 (blue triangles), and serum-deprived and treated with 50 μM VX-765 (pink triangles) neurons. **d** Secreted or cellular $A\beta_{42}/A\beta_{38} + A\beta_{40} + A\beta_{42}$. **e** Secreted Il-1β ($F_{(3,8)} = 6.636$, $p = 0.0146$), IFN-γ, TNF-α ($F_{(3,8)} = 3.931$, $p = 0.054$), and IL-6 ($F_{(3,8)} = 10.24$, $p = 0.0041$) relative to untreated (+Serum) HPN. ANOVA, Dunnett's post-hoc versus +Serum, $*p<0.05$. **a, c, d, e**. Each individual neuron preparation tested is represented by one symbol. Data represent mean and s.e.m.

AD-like pathologies[5]. LTP and dendritic spine density are restored, and increased Aβ levels are reduced, in *APP/PS1/Casp1*$^{-/-}$ mice[5]. While the defect in AD immunity is largely attributed to CNS resident microglia, astrocytes and perivascular myeloid immune cells, we have identified abnormal activation of the Nlrp1 inflammasome in AD neurons, resulting in Casp1 activation, which then can activate Casp6-mediated neurodegeneration and Il-1β-mediated glial inflammation[11]. By targeting Casp1 in both neurons and glia, VX-765 provides a double protection against these coexisting degenerative AD features.

The reduction of Aβ levels in VX-765-treated mice was unexpected. After only 12 treatments over a 3-month span, RIPA- and formic acid-soluble, immunostained, and thioflavin S-positive Aβ levels were considerably lower than in vehicle-treated J20 mice and remained similar to levels in pretreated 5-month-old J20 brains. Similarly, the absence of *Casp1* in the J20/*Casp1*$^{-/-}$ decreased Aβ deposits in the brain. These results indicate that VX-765 and the absence of *Casp1* stop the progressive

accumulation and deposition of Aβ. Furthermore, it became apparent that cremophor vehicle injections increased Aβ accumulation in the J20 mouse of the VX-765 study compared to the *Casp1* null study. It is tempting to speculate that either the cremophor or the injections mimic increased inflammation in the aged brain.

Since VX-765 inhibits inflammatory Casp1, the results indicate that in the J20 mouse model, inflammation may be driving Aβ accumulation and contrast with the more popular view that Aβ drives inflammation (see Supplementary Fig. 10). These results are consistent with recent evidence indicating that microglia elimination prevents amyloid deposition[3,4], and increased peripheral inflammatory markers in midlife humans are associated with brain atrophy and decreased cognition in ageing[43].

Whereas the J20 is a familial AD mouse model, the presence of the Nlrp1-Casp1-Casp6 pathway in mild cognitively impaired and in the early and late stages of sporadic human AD suggests that the treatment will also work in sporadic AD. VX-765's

selectivity for Casp1[25,26], nontoxicity to human neurons and tolerability in phase 2b clinical trials against epilepsy[29], blood–brain barrier permeability, and prevention of neurode-generation in cultured human CNS neurons suggest that it is a most promising novel therapy against both sporadic and familial forms of AD.

It is likely that an efficient treatment against AD will require very early treatment to prevent neuronal degeneration. Similar to antihypertensive or anticholesterol medications that protect against stroke or heart infarcts, we believe that the brain requires preventative early treatments before serious damage occurs. With this in mind, we chose to treat J20 mice only 1 month after the onset of cognitive decline. The outcome was unexpectedly positive. These results suggest that an efficient therapeutic intervention with VX-765 is likely to be more successful in mild cognitively impaired individuals or at the very early onset of clinically diagnosed AD.

## Methods

**Study design**. *VX-765 study*: Five-month old vehicle-injected WT (WT + vehicle; n = 18) and J20 mice (J20 + vehicle: n = 14), and VX-765-injected J20 mice (J20 + VX-765: n = 13) mice were longitudinally assessed at five different time points: baseline before treatment (baseline untreated J20 were grouped together; n = 19), after three IP injections per week of VX-765 or vehicle (Treatment 1), after six additional injections over 2 weeks (Treatment 2), after 4-week washout period (WO), and after an additional three injections in 1 week (Treatment 3) (Fig. 1a). All animals tested were included in the analyses unless (a) the animal died during the experiment, or (b) the animal did not behaviourally respond. A total of 25 litters, spread across four different cohorts and tested at different times, were used. Three of these cohorts underwent NOR and open field testing, while cohort 4 was used for Barnes maze. After animals were baseline tested to confirm that J20 animals were behaviourally deficit, J20 animals were randomly assigned to either vehicle or VX-765 treatment independent of behavioural performance. Of all animals used, four J20 mice showed no behavioural deficits at the beginning of the experiment and were not used. Two J20 + vehicle mice did not move at all during the experiment and their behavioural data excluded; however, these animals were kept and were still used for post-mortem analysis.

*Casp1 validation study*: To validate Casp1-specific effects of VX-765, J20 mice were generated on a Casp1[−/−] background. Seventy-three mice were used (n = 12 −13 per genotype, six different genotypes), all of which were bred through in vitro fertilization (IVF) from 35 donor females. Breeding details are described in the Animal Studies section below. All mice were behaviourally assessed between 7 and 8 months of age, and no animals were excluded from this study.

All mice were sacrificed and processed at 8 months of age. Behavioural scoring was blinded to mouse genotype and treatment, and all animals were randomly assigned for either biochemical or histological analysis and blindly analysed.

**VX-765, VRT-043198 IC$_{50}$ assays on recombinant caspases**. Recombinant caspases were commercially obtained (Biovision, Milpitas, CA, USA; K233-10-25) and used according to the manufacturer's protocol. Caspase activity was measured using a fluorescence-based assay using the following substrates (Enzo Life Sciences, NY, USA): N-Ac-Tyr-Val-Ala-Asp-7-amino-4-trifluoromethylcoumarin (Ac-YVAD-AFC) for human Casp1 and mouse Casp1, N-Ac-Val-Asp-Val-Ala-Asp-7-amino-4-trifluoromethylcoumarin (Ac-VDVAD-AFC) for human Casp2, N-Ac-AspGlu-Val-Asp-7-amino-4-trifluoromethylcoumarin (Ac-DEVD-AFC) for human Casp3 and Casp7, Ac-Trp-Glu-His-Asp-7-amino-4-tri-fluoromethylcoumarin (Ac-WEHD-AFC) for human Casp4 and Casp5, and mouse Casp11, N-Ac-Val-Glu-Ile-Asp-7-amino-4-trifluoromethylcoumarin (Ac-VEID-AFC) for human Casp6, Ac-Ile-Glu-Thr-Asp-7-amino-4-trifluoromethylcoumarin (Ac-IETD-AFC) for human Casp8 and Casp10, and N-Ac-Leu-Glu-His-Asp-7-amino-4-trifluoromethylcoumarin (Ac-LEHD-AFC) for human Casp9. VX-765 (100 μM–100 pM) and VRT-043198 (100 μM–1 pM) (DSK Biopharma, Morrisville, NC) were mixed with 0.25 units of caspase and 10 μM substrate in a buffer containing 50 mM 4-(2-hydroxyethyl)-1-piperazineethanesulfonic acid (HEPES; pH 7.2), 50 mM NaCl, 0.1% 3-[(3-cholamidopropyl)dimethylammonio]-1-propane-sulfonate (CHAPS), 10 mM ethylenediaminetetraacetic acid (EDTA), 5% glycerol, and 10 mM dithiothreitol. Enzyme activity was measured in a Synergy H4 plate reader (BioTek, Winooski, VT, USA) at 380 nm excitation, 505 nm emission wavelengths every 30 s for 30 min at 37 °C. Cleavage rates were calculated from the linear phase of the assay. Fluorescence units were converted to the molar concentration of released 7-amino-4-trifluoromethylcoumarin (AFC) based on a 0–25.0 μM free AFC standard curve. VX-765 IC$_{50}$ values were determined by plotting relative activity (% DMSO control) to inhibitor concentrations. Data were fitted to a dose−response curve equation containing a −1 slope with top and bottom constraints of 1 and 0, respectively.

**Animals studies**. All animal procedures followed the Canadian Council on Animal Care guidelines and were approved by the McGill University Animal Care committee. The J20 transgenic mouse line (JAX Stock No. 006293, Jackson Laboratories, Bar Harbor, ME, USA) expresses the Swedish (670/671$_{KM→NL}$) and Indiana (717$_{V→F}$) human *APP* mutations under the PDGF-βpromoter. Male J20 mice were used as breeders and their sperm was used for IVF colony expansion.

The J20/Casp1 cohort was generated following the animal breeding standard protocols of the Institut de recherche en immunologie et en cancérologie (IRIC) at the Université de Montréal (Montreal, QC, Canada). Casp1 null mice were obtained from Jackson Laboratories (B6N.129S2-Casp1$^{tm1Flv}$/J). IVF was carried out using fresh sperm from J20/Casp1$^{+/−}$ males and the eggs of WT/Casp1$^{+/−}$ donor females. Fertilized eggs were reimplanted in 20 C57Bl/6 females, generating pups with six possible genotypes: WT/Casp1$^{+/+}$ (WT/WT), WT/Casp1$^{+/−}$ (WT/Het), WT/Casp1$^{−/−}$ (WT/KO), J20/Casp1$^{+/+}$ (J20/WT), J20/Casp1$^{+/−}$ (J20/Het), J20/Casp1$^{−/−}$ (J20/KO). Both males and females were used, aged at the IRIC, and then transferred to our laboratories prior to the start of experiments.

Genotypes were determined by tail biopsy and RT-PCR. Mice were group-housed (2−4 animals per cage) in standard macrolon cages under a 12-h reverse light/dark cycle and controlled environmental conditions. Food and water were available ad libitum.

VX-765 (Adooq Bioscience, Irvine, CA) was dissolved in 20% cremophor in dH$_2$O (Sigma-Aldrich, Oakville, ON, Canada) and administered intraperitoneally. Vehicle-treated J20 and WT mice received cremophor only.

**Blood–brain barrier permeability analysis**. Five-month-old J20 and WT mice were anesthetised with isoflurane and the right carotid artery was separated. A small incision was made along the carotid wall and a micro-catheter was inserted into the entrance of the internal carotid artery. The catheter was secured in place using suture thread. VX-765 (50 mg kg$^{−1}$) was infused at 50 μl min$^{−1}$ (90–120 μl total vol). The micro-catheter was left an additional 5 min to allow diffusion, after which blood was collected by intra-cardiac puncture. The mouse was trans-cardially perfused with ice-cold saline and the hippocampus and cortex were dissected out. Samples were sent to the Biopharmacy platform (Université de Montreal) for liquid chromatography and tandem mass spectrometry analysis.

**Behavioural analysis**. *Open field*: locomotor activity was measured using an HVS2100 automated tracking system (HVS Image, Hamptom, UK). Mice were placed in the open field chamber (40 × 40 cm Plexiglas box, no ceiling, and white floor) and allowed to explore for 5 min.

*Novel object recognition (NOR)*: mice were pre-exposed to two identical objects in the open field chamber for 5 min. Two hours after pre-exposure, mice were placed back into the chamber and exposed to one familiar object and one novel object for 5 min. Mice were exposed to a specific object only once across different test sessions, and novel object placement was counterbalanced within each treatment group. Percentage touches of the novel object during the test phase and discrimination index ((# touches novel − # touches familiar)/total touches) were assessed.

*Barnes maze*: the Barnes maze platform (91 cm diameter, elevated 90 cm from the floor) consisted of 20 holes (each 5 cm in diameter). All holes were blocked except for one target hole that led to a recessed escape box. Spatial cues, bright light, and white noise were used to motivate the mice to find the escape during each session. For the adaptation phase, each mouse explored the platform for 60 s. Any mouse that did not find the escape box was guided to it and remained there for 90 s. For the acquisition phase, each trial followed the same protocol, with the goal to train each mouse to find the target and enter the escape box within 180 s. Mice remained in the box for an additional 60 s. Four trials per day, approximately 15 min apart, were performed for 4 consecutive days. In the probe test (day 5), each mouse performed one 90 s trial. The target was still located in the same position, but the escape box was blocked. The latency and errors to reach the target, plus the number of mouse pokes to each of the holes (target preference) were measured. The Y-Maze was composed of a symmetrical Y-shaped maze with three arms 120° from each other, designated A, B, and C. Animals were placed at the distal end of arm A and allowed to explore the maze for 5 min. Total arm entries and spontaneous alternation, defined as consecutive entries in three different arms, were recorded. Percentage alternation [(# alternations/total number arm entries − 2) × 100] was determined.

**Tissue processing**. For immunohistochemistry, mice (n = 4−6 per group) were anesthetised with isofluorane and transcardially perfused with ice-cold saline followed by ice-cold 4% paraformaldehyde (Sigma-Aldrich, St Louis, MO, USA) in 0.1 M phosphate-buffered saline. Brains were post-fixed in 10% neutral-buffered formalin (Thermo Fisher Scientific, Mississauga, ON, Canada) for 16 h and transferred to 70% ethanol. Brains were paraffin embedded and sectioned at 4 μm.

For western blot and ELISA (n = 4−8 per group), anesthetised mice were cervically dislocated, the hippocampus and cortex dissected out and immediately frozen on dry ice. Proteins were extracted in 5× volume/weight with radioimmunoprecipitation assay (RIPA) buffer (50 mM Tris-HCl, pH 7.4, 1% NP-40, 150 mM NaCl, 0.25% Na-deoxycholate, 1 mM EDTA, 1 mM Na$_3$VO$_4$, 1 mM NaF, 1 mM PMSF, 10 μl ml$^{−1}$ of Proteases Inhibitors Cocktail (Sigma-Aldrich)) on

ice with a tissue homogenizer (OMNI International, Kennesaw, GA, USA). Samples were centrifuged (20,000 × g, 4 °C) for 20 min, the supernatant recovered, and protein quantified using the BCA method. The RIPA-insoluble pellet was further extracted in 70% formic acid in $dH_2O$, evaporated, and resolubilized in 200 mM Tris-HCl, pH 7.5.

**Immunohistological staining**. Epitopes were demasked in either citrate (10 mM Tris-Na Citrate, pH 6.0) or EDTA (10 mM Tris Base, 1 mM EDTA, 0.05% Tween-20, pH 9.0) antigen retrieval buffer, and immunostained using the Dako Auto-stainer Plus automated slide processor and EnVision Flex system (Dako, Bur-lington, ON, Canada). Following deparaffinization and rehydration, sections were peroxidase treated, blocked in serum-free protein block, and immunostained with the following antibodies diluted in EnVision Flex Antibody Diluent: 1:2000 rabbit anti-Iba1 (Wako 019-19741, Richmond, VA, USA), 1:8000 rabbit anti-GFAP (Dako Z-0334), 1:2000 rabbit anti-A$\beta_{1-40}$ (F25276, laboratory developed), and 1:1000 mouse antisynaptophysin (Sigma-Aldrich S5768). Sections were treated with the kit-supplied appropriate mouse or rabbit HRP-conjugated secondary antibody and visualized with DAB. Sections were hematoxylin counterstained, dehydrated, and coverslipped with Permount (Fisher Scientific). Sections were digitally scanned with MIRAX SCAN for analysis (Zeiss, Don Mills, ON, Canada). After deparaffinization and rehydration, slides were stained with filtered 1% aqueous Thioflavin-S (Sigma-Aldrich).

**Quantitative immunohistological analysis**. Iba1-positive cells in the cortex and anterior portion of the CA1 region of the hippocampus were quantified using a modified version of the areal counting frame[48]. Briefly, every fifth slide was quantified (resulting in four sections per brain). Multiple ×80 images were taken with the MIRAX SCAN within the area of interest and the number of Iba1-positive cells were manually counted. The numerical density (number of cells mm$^{-3}$) in the anterior CA1 region and cortex was estimated. A minimum number of 150 hits per area were needed to make a reliable estimate. Additional sections were counted if we were unable to reach our minimum number. Quantification was performed blinded for treatment and genotype. ImageJ software (NIH, Bethesda, MD, USA) was used to measure Aβ accumulation, GFAP and synaptophysin immunopositive staining in the CA1 region and cortex. Four sections per brain were analysed, and multiple images were taken within each section to cover the CA1 region and cortex. Threshold was equally adjusted across all brains to highlight the stained area, and particles were analysed to determine total area of staining. Results are expressed as total area stained per total area analysed ($\mu m^2$ mm$^{-2}$). Representative images have only been cropped to conform to size constraints and have not undergone any post-processing.

**Western blotting**. Mouse protein samples (25–100 µg) were prepared in Laemmli (5% 2-β-mercaptoethanol, 1 mM $Na_3VO_4$, 1 mM NaF, 1 mM PMSF, 10 µl ml$^{-1}$ of Proteases Inhibitors Cocktail (Sigma-Aldrich)), boiled for 5 min, and separated by polyacrylamide gel electrophoresis (PAGE). Samples were probed with the fol-lowing primary antibodies, diluted in either 5% nonfat dry milk in Tris-buffered saline and 0.1% Tween-20 or PBS blocking buffer (Thermoscientific): 1:1000 mouse anti-human A$\beta_{1-16}$ (6E10) (BioLegend 803001, San Diego, CA, USA), 1:2000 rabbit anti-APP C-terminus (Sigma-Aldrich A8717), 1:1000 rabbit anti-neprilysin (Abcam ab79423), 1:500 rabbit anti-IDE (Abcam ab32216), 1:1000 rabbit anti-Caspase-3 (Cell Signaling 9665), 1:1000 mouse antiactive Caspase-8 (Cell Signaling 8592), 1:1000 rabbit anti-Caspase-9 (Cell Signaling 9504), and 1:5000 mouse anti β-actin (Sigma-Aldrich A5441). Blots were detected with 1:5000 HRP-linked anti-mouse (GE Amersham NA9310, Montreal, QC, CA) or 1:5000 anti-rabbit (Dako P0217) secondary antibody followed by ECL (GE Amersham). Immunoreactive bands were visualized using ImageQuant LAS 4000 imaging system (Fujifilm USA, Valhalla, NY, USA), and densitometric analyses were per-formed with Image Gauge analysis software (Fujifilm USA). Brightness/contrast of raw blots were equally adjusted across the entire image with Adobe Photoshop CS5 software (Adobe Systems, Ottawa, ON, CA) to generate representative images. No additional modification was made. Canvas$^{TM}$ X software (Acd system, Plan-tation, FL, USA) was used to create final figures. Raw blots for western blots are available in Supplementary Figure 11.

**ELISA**. Pro- and anti-inflammatory cytokines, and Aβ levels in mouse brain were determined using electrochemiluminescence immune assay kits from Meso Scale Discovery (Rockville, MD, USA). Standards and samples were prepared according to the manufacturer's protocols and run in duplicate. The ten-plex mouse proin-flammatory panel was used for mouse hippocampal and cortical and samples. A single Il-1β kit was used to measure mouse plasma. The four-plex human proin-flammatory panel was used for HPN samples (see below). The four-panel human Aβ 6E10 kit was used for mouse hippocampus and cortex, and HPN samples.

**RNA extraction and cDNA synthesis**. Total RNA was extracted from mouse hippocampus and cortex (n = 3 per group) with Qiazol (Qiagen, Valencia, CA, USA) followed by homogenization with a 20-gauge needle. The miRNeasy mini kit, including on-column DNase treatment step, was used to purify total RNA (Qia-gen). RNA quality and quantity were determined using a spectrophotometer (DS-

11 FX+, DeNovix, Wilmington, DE, USA). Five hundred nanograms of total RNA was reverse transcribed for each sample (with avian myeloblastosis reverse tran-scriptase (AMV-RT)) (Roche, Mannheim, Germany).

**PCR and real-time PCR**. HAPP transgene PCR amplification was performed using *Taq* DNA polymerase (New England Biolabs, Whitby, ON, Canada) using the following primers: (hAPP) 5′-AACACAGAAAACGAAGTT-3′ and 5′-CCGATG GGTAGTGAAGCA-3′ (480-bp amplicon)[31], (Hypoxanthine-guanine phosphor-ibosyltransferase, *HPRT*) 5′-GTAATGATCAGTCAACGGGGGAC-3′ and 5′-CCA GCAAGCTTGCAACCTTAACCA-3′ (177-pb amplicon) (Carcinogenesis). Pro-ducts were visualized on RedSafe (FroggaBio, North York, ON, Canada) stained 2% agarose gels. Real-time PCR experiments were performed with SYBR Green Taq Mastermix (Quanta BioSciences, Gaithersburg, MD, USA) on an Applied Biosys-tems 7500 Fast Real-Time PCR system machine (Applied Biosystems, Foster City, CA, USA). Gene amplification was performed with the following primers: (18s) 5′-GTAACCCGTTGAACCCCAT-3′ and 5′-CCATCCAATCGGTAGTAGCG-3′; (IDE) 5′- ACTAACCTGGTGGTGAAG-3′ and 5′- GGTCTGGTATGGGAAATG -3′; (Neprilysin) 5′- TCTTGTAAGCAGCCTCAGCC-3′ and 5′- CTCCCCACAG CATTCTCCAT-3′. Results are expressed as fold-induction values normalized to mean HPRT and 18S reference genes using the Pfaffl's method.

**Mouse synaptic plasticity RT$^2$-Profiler PCR array**. The mouse synaptic plasticity RT$^2$ Profiler PCR Array (PAMM-126Z, Qiagen) profiles the expression of 84 genes involved in synaptic alterations during learning and memory and five housekeeping genes (β-actin; β-2-microglobulin; glyceraldehyde 3-phosphate dehydrogenase, GAPDH; β-glucuronidase, Gusb; Heat shock protein 90 alpha (cytosolic), Hsp90ab1) with real-time PCR. Total RNA (465 ng for each sample) was reverse transcribed (which included a genomic DNA elimination step) following the manufacturer's protocol (First Strand cDNA Synthesis Kit, Qiagen). The real-time PCR reaction was performed in a real-time cycler ABI 7500 (Fast Block) (Applied Biosystems) using the RT$^2$ SYBR Green/ROX PCR master mix (Qiagen). Cycling conditions were as follows: 2 min at 50 °C, 10 min at 95 °C, 40 cycles, 15 s each at 95 °C, and 1 min at 60 °C. Threshold and baseline were set manually according to the manufacturer's instructions. Data were normalized to the geometric mean of the five housekeeping genes and analysed by the comparative Ct-method ($2^{-\Delta CT}$).

**Neuronal cell culture and axonal degeneration assay**. Twelve- to sixteen-week-old fetal cortical tissue were obtained from the Birth Defects Research Laboratory (University of Washington, Seattle, USA) in accordance with an approved protocol by the McGill institutional review board. HPN were cultured from brains that were dissociated with trypsin, treated with deoxyribonuclease I, and filtered through 130, 70, and 30 μm nylon mesh[21]. Dissociated cells were centrifuged for 10 min at 300 × g and resuspended in minimal essential medium (MEM) with Earle's salts, and supplemented with 5% decomplemented BCS, 1× Penicillin-Streptomycin, 1 mM sodium pyruvate, and 2 mM L-glutamine (Invitrogen, Carlsbad, CA, USA). Cells were seeded at a density of 3×10$^6$ cells ml$^{-1}$ on 5 µg mL$^{-1}$ poly-L-lysine-coated six-well tissue culture plates (Sigma-Aldrich). MEM medium was changed every 2 days and astrocyte proliferation was limited with fluorodeoxyuridine (FdU) antimitotic treatment (1 mM, Sigma) for the first 4 days. Neuronal cultures were grown 7 −10 days before conducting experiments. Four different neuron preparations at different times were tested. One of the neuron preparations was not stable and the culture, including the Serum + control that did not receive any drug, died in the middle of the experiment. Therefore, three different neuron preparations from three genetically different donors were used.

VX-765 toxicity was assessed with 3-(4,5-dimethylthiazol-2-yl)-2,5-diphenyltetrazolium bromide (MTT). Neurons were treated with 25–200 μM VX-765 or 0.1% DMSO (highest DMSO concentration used) for 72 h. Neurons were then incubated with 0.5 µg ml$^{-1}$ MTT (Sigma-Aldrich) for 4 h at 37 °C (5% $CO_2$). The formazan crystals were dissolved in 100 µl DMSO for 30 min. Absorbance was measured at 560 and 670 nm using the Synergy H4 plate reader.

Axonal degeneration was induced either by APP$^{WT}$ transfection or serum deprivation stress. VX-765 was administered either as a pretreatment strategy (1 h prior to stressor) or as a postbeading treatment strategy (48 h after stressor). Neurons were transfected by Gene gun (BioRad, Mississauga, ON, Canada) with gold beads coated with pBudCE41-eGFP for fluorescence visualization. Neurons that were APP$^{WT}$ stressed were transfected with pBudCE41-eGFP/APP$^{WT}$. Briefly, neurons were treated with medium supplemented with 25 or 50 μM VX-765, 5 μM Casp1 inhibitor Z-YVAD-fmk, or DMSO. Fluorescence images were taken every 24 h (up to 72 h post-treatment) with a Nikon Eclipse Ti microscope (Nikon, Mississauga, ON, CA) and Photometrics coolSNAP HQ2 CCD camera (Photometrics, Tucson, AZ, USA) using NIS Elements AR 3.10 software (Nikon). Percentage neuronal beading was measured by counting the number of beaded eGFP-positive neurons over total number of eGFP-positive neurons at each time point. A minimum of 150 eGFP-positive neurons was counted for each condition. Conditioned medium was sampled (supplemented with 1 mM $Na_3VO_4$, 1 mM NaF, 1 mM PMSF, 10 µl ml$^{-1}$ of Proteases Inhibitors Cocktail (Sigma-Aldrich)), and neurons were harvested in cell lysis buffer (50 mM HEPES, 0.1% CHAPS, 0.1 mM EDTA) for analysis.

**Statistical analyses**. The number of animals or independent experiments, and statistical analysis used (ANOVA and post-hoc comparisons) are all indicated in the figure legends. All values are expressed as the mean ± s.e.m., with F and p values indicated in the figure legends. All statistical analyses were conducted using GraphPad Prism 7 software (GraphPad Software, La Jolla, CA, USA).

## Data availability

For data from individual mice tested in behavioural tests, see the Supplementary section. Digital scans of immunohistological staining have been saved electronically and can be made available upon request and provision of a depository site with sufficient memory to accept the files. Any additional information is available upon request.

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

## Acknowledgements

We are indebted to Edith Hamel for sharing her laboratory for the blood–brain barrier experiments. We would like to thank the animal quarters staff, especially Véronique Michaud, for maintaining the mice, the IRIC Université de Montréal pathology core for embedding and cutting brain tissue sections. A.N. received a postdoctoral scholarship from the Alzheimer Society of Canada. This work was supported by funds from the Canadian Institutes of Health Research (CIHR) MOP-243413-BCA-CGAG-45097, CIHR project grant 153097, and the JGH Foundation to A.C.L.

## Author contributions

J.F.: animal behaviour and analyses, immunohistochemistry analyses, ELISA, cell culture experiments, blood–brain barrier permeability assays, designed experimental paradigm, made figures, wrote methods, results and revised manuscript. B.F.: molecular biology, immunostaining, prepared figures, wrote methods and results, revised the manuscript. A.N.: western blot analyses, prepared figures, wrote methods and results, revised the manuscript. J.L.: performed experiments for $IC_{50}$ on recombinant enzymes, wrote methods and results. Revised the manuscript. C.L.: performed blood–brain barrier permeability experiments, revised the manuscript. A.C.L.: Conception of the experimental idea and design, supervision of data collection and analysis, wrote manuscript.

## Additional information

**Competing interests:** The authors declare no competing interests.

