## [Peer Review File · Nature Communications]

Reviewers' comments:

Reviewer #1 (Remarks to the Author):

The authors have prepared a manuscript using cultured cells and live mice to investigate the effects of the caspase-1 inhibitor VX-765 on behavior, inflammation, Abeta deposition and neuritic dystrophy. Using J20 mice, they show that acute administration of VX-765 reverses preexisting behavioral deficits, reduces inflammation and Abeta deposition. Using cultured human neurons, they show reduced neuritic dystrophy, Abeta levels and inflammatory markers. It is remarkable how quickly the effects of VX-765 were observed in J20 mice. These are interesting and important results, providing strong preclinical support for VX-765 as a potential treatment for Alzheimer's disease.

The data were acquired using an impressive variety of techniques. The following concerns pertain to exclusion of mice, group sizes, analysis of the data and interpretation of the data.

- 1) Indicate the treatment and transgene status of the mice that were excluded from the analyses of the behavioral data, and whether there was a bias for excluding poor-performing VX-765 treated or J20/Casp1^{-/-} mice.
 - 2) Indicate whether treatment was randomized across litters.
 - 3) Group sizes for behavioral results in Supplementary Figure 4 and mRNA levels in Figure 4 are probably too small to be reliable (N = 2-3 in some groups) and therefore subject to beta- and alpha-type errors.
 - 4) Figure 4 data for Abeta comparing baseline (5 month-old) and treated or untreated mice (8 month-old) show no increases and in some instances even decreases in the older, untreated mice that have not been reported by other researchers using J20 mice, calling into question the reliability of the results. In addition, APP levels in the western blots appear to be higher in the treated than untreated mice. Is this real?
 - 5) Supplementary Figure 1 shows IC50 values in the nanomolar range for not only caspase-1 but also caspase-5, -8, -9 and -10. The same figure shows concentrations of VX-765 in the hippocampus and cortex ranging from ~300 nM to 11,000 nM; thus, treatment could potentially inhibit other caspases besides caspase-1. Were other caspases inhibited in the experimental mice?
 - 6) Supplementary Figure 4 shows no apparent decrease in Iba1 staining between J20/Casp1^{+/+} and J20/Casp1^{-/-} mice, in contrast to J20 mice treated with VX-765, arguing against the effects of VX-765 being solely due to inhibition of Casp1.
 - 7) Supplementary Figure 6 shows photomicrographs of Abeta staining in J20/Casp1^{+/+} and J20/Casp1^{-/-} mice, but the relative levels of deposition are hard to gauge. Plaque load measurements would help differentiate background staining from plaque deposition.
 - 8) Some data in the current paper may contradict the conceptual model presented in Supplementary Figure 7. In principle, the inflammasome pathway (Nlrp1→Casp1→Casp6) is blocked in Casp1^{-/-} mice. So, if inflammation drives plaque formation, then the J20/Casp1^{-/-} mice should not have plaques. However, this is not the case, as illustrated in Supplementary Figure 6. One wonders if VX-765 has pleotropic effects – acting not only on Casp1 (and possibly other caspases) but also on APP processing and/or Abeta clearance (Figures 4h and 4i notwithstanding, as the groups sizes are rather small). This situation would not detract from the description of the effects of VX-765, but has implications in understanding how the treatment works.
- Overall, this paper describes beautifully performed experiments showing remarkable effects of VX-765 on behavior, inflammation and amyloid deposition in J20 mice. The validity of the conclusions would be strengthened, however, by addressing the issues delineated above or by modifying the conclusions.

Reviewer #3 (Remarks to the Author):

In this manuscript, Flores et al. show in vivo proof of concept for use of VX-765, a caspase-1 inhibitor, as a treatment for Alzheimer disease (AD) in the J20 familial AD mouse model. They present thorough in vitro selectivity data to demonstrate that VX-765 selectively inhibits caspase-1 in human and mouse and demonstrate that this compound rescues cognitive and memory deficits and attenuates hyperactivity in the J20 mouse model. These effects are shown to be dependent on VX-765 dose, and their dependence on caspase-1 inhibition was demonstrated in a caspase-1 null mouse. Further, VX-765 was shown to reduce markers of neuroinflammation, to inhibit the accumulation of A β , and to restore levels of synaptophysin. In human primary neuron cultures, VX-765 was demonstrated to prevent human neuronal degeneration on the basis of its ability to inhibit neuritic beading. Taken together, these results represent a convincing in vivo proof of concept for caspase-1 inhibition by VX-765 as a treatment for AD. Based on the absence of treatments that delay or reverse cognitive deficits associated with AD, this manuscript represents a significant advance and is well suited for publication in Nature Communications with minor revisions.

Comments:

1. It is interesting that VX-765 attenuates hyperactivity in J20 mice, but that hyperactivity persists in the caspase-1 null mouse. Can the authors comment on this?
2. In general, the accessibility of the paper to a general audience could be improved. For example, the significance of thioflavin S and Iba1 staining could be explained briefly in the results section, and more explanation of which hypothesis each experiment is designed to test could be given.

We thank the reviewers for their insightful comments (**in bold**) and have now revised the paper accordingly, as described in our response to every comment below.

Reviewer #1 (Remarks to the Author):

The authors have prepared a manuscript using cultured cells and live mice to investigate the effects of the caspase-1 inhibitor VX-765 on behavior, inflammation, Abeta deposition and neuritic dystrophy. Using J20 mice, they show that acute administration of VX-765 reverses pre-existing behavioral deficits, reduces inflammation and Abeta deposition. Using cultured human neurons, they show reduced neuritic dystrophy, Abeta levels and inflammatory markers. It is remarkable how quickly the effects of VX-765 were observed in J20 mice. These are interesting and important results, providing strong preclinical support for VX-765 as a potential treatment for Alzheimer's disease. The data were acquired using an impressive variety of techniques. The following concerns pertain to exclusion of mice, group sizes, analysis of the data and interpretation of the data.

1) Indicate the treatment and transgene status of the mice that were excluded from the analyses of the behavioral data, and whether there was a bias for excluding poor-performing VX-765 treated or J20/Casp1^{-/-} mice.

The J20 mice are tested before the VX-765 treatment to ensure that they exhibit behavioural deficits. Out of a total of 83 J20 animals used in these experiments, 4 J20 mice were excluded at the beginning of the experiment since they did not show deficits in NOR. Two J20 + Vehicle animals were excluded from behavioural analysis. One of the J20 + Vehicle animals was part of our dose-response experiment and did not move during Treatment 2 behavioral testing (Open Field + NOR). The other J20 + Vehicle animal was part of our main study and did not move during Open Field at Treatment 1 and Treatment 2. This information was added to the first paragraph of the Methods section.

2) Indicate whether treatment was randomized across litters.

All animals were randomized across litters. This information was added to the first paragraph of the Methods study design section.

3) Group sizes for behavioral results in Supplementary Figure 4 and mRNA levels in Figure 4 are probably too small to be reliable (N = 2-3 in some groups) and therefore subject to beta- and alpha-type errors.

We only had preliminary results for the J20/Casp1^{-/-} when we submitted the paper. Anticipating that this would be a problem, we bred more animals and obtained 8 month old J20/WT, J20/Casp1^{+/-}, and J20/Casp1^{-/-} as the review came in. We now have studied n=12 WT/WT for behaviour; and n=8 each for biochemistry and histology, n=13 WT/Casp1^{+/-} for behaviour; and n=10 each for biochemistry and histology, n=12 WT/Casp1^{-/-} for behaviour; and n=12 for biochemistry and n=10 for histology, n=12 J20/WT (Casp1^{+/+}) for behaviour; and n=6 each for biochemistry and histology, n=12 J20/ Casp1^{+/-} for behaviour; and n=10 each for biochemistry and histology, n=12 J20/ Casp1^{-/-} for behaviour; and n=13 for biochemistry and n=12 for

histology. The behavioral data has been added as new figure 3, inflammation data in Figure 4, and A β /APP data in Figure 5 of the manuscript. Additional data assessing A β , APP and CTF, IDE, and NEP has been added in Supplementary Fig. 7 and 8.

We had enough naïve J20 to repeat VX-765 treatments on n=4 to 5 animals per group. Additional data with ELISA increased the n to 7 to 8.

Regarding APP mRNA levels in old Figure 4, we opted to test the levels of APP protein with two different antibodies; the human specific 6E10 and the anti-mouse/human APP C-terminal antibody since protein levels reflect more appropriately the production of A β . We now show in Figure 5 (6E10) and Supplementary Fig. 7 (anti-APP C-terminal) the results of n=6 to 7 samples/condition. These results clearly show that there is no effect of either VX-765 treatment or the absence of Casp1 on APP or APP CTF levels.

Regarding IDE and NEP, we also opted to use the tissues for protein analyses (Supplementary Fig. 8). Again, the results show that there is no effect of either VX-765 treatment or the absence of Casp1 on IDE or NEP protein levels.

4) Figure 4 data for Abeta comparing baseline (5 month-old) and treated or untreated mice (8 month-old) show no increases and in some instances even decreases in the older, untreated mice that have not been reported by other researchers using J20 mice, calling into question the reliability of the results.

Our additional analyses do not significantly change our results and interpretation. Differences with previous studies may reflect the use of the MSD multiplex ELISAs measuring A β _{38, 40, and 42} compared to another type of ELISA.

Nevertheless, we did note an effect of cremophor vehicle injections in exacerbating the levels of immunohisto A β in 8 month old mice hippocampus and cortex (compare Fig. 5b J20+veh versus J20/WT in Fig. 5c). This could explain variable results with those reported.

Comparison of RIPA-soluble A β levels in 5 month old J20 Baseline (untreated) in the VX-765 study and 8 month old J20/WT in the Casp1 study indicates higher levels of A β ₄₂/total A β in the 8 month old hippocampus and cortex. Total RIPA-soluble A β levels do not change between 5 to 8 month old hippocampus but are doubled in 8 month old cortex compared to 5 month old cortex. Formic acid levels of total or A β subtypes did not seem to change significantly.

In addition, APP levels in the western blots appear to be higher in the treated than untreated mice. Is this real?

No. We have repeated these experiments with additional samples and with two different antibodies; the 6E10 human specific antibody and the mouse/human anti-APP C-terminal antibody (Sigma). The results (Fig. 5 l,m and Supplementary Fig. 8c&d) show similar levels of full length APP in untreated, vehicle-treated and VX-765-treated mice. We also measured these in the Casp1 null study and find no change in APP levels (Fig. 5n,o and Supplementary Fig.

8e,f). Furthermore, we measured the CTF in both studies and find no change (Supplementary Fig. 8c-f)

5) Supplementary Figure 1 shows IC50 values in the nanomolar range for not only caspase-1 but also caspase-5, -8, -9 and -10. The same figure shows concentrations of VX-765 in the hippocampus and cortex ranging from ~300 nM to 11,000 nM; thus, treatment could potentially inhibit other caspases besides caspase-1. Were other caspases inhibited in the experimental mice?

The reviewer is correct. Given the potency of VX-765 for these caspases, it would be possible that VX-765 would display non-Caspase-1 effects. Caspase-5 and caspase-10 can be excluded here because they do not exist in mice (Reed et al., Genome research 2003). We evaluated the IC50 of VX-765 on mouse caspase-11 since it is considered the mouse caspase-4 and -5. The IC50 was in μ M concentration so mouse caspase-11 is unlikely to be affected by VX-765.

Remaining possibilities were Caspase-8 and Caspase-9. VX inhibits Casp8 and 9, at 149 nM and 37.3 nM, respectively. VRT inhibits Casp8 and 9, at 142 nM and 54.8 nM, respectively. To assess the possibility that VX-765 inhibits these caspases in vivo, we analysed the abundance of the active subunit of caspase-8 in the spleen of the mice. Caspase-8 is naturally active in these tissues. Spleens from vehicle or VX-765-treated mice did not show a difference in the level of active Caspase-8 subunits present in the spleens. Furthermore, we evaluated Casp9, active Casp8, and Casp3, since it is activated by Casp8 and Casp9, in hippocampus and cortex (New Supplementary Fig. 9). No significant inhibition of these three caspases was observed in the hippocampus and cortex after VX-765 treatment.

6) Supplementary Figure 4 shows no apparent decrease in Iba1 staining between J20/Casp1^{+/+} and J20/Casp1^{-/-} mice, in contrast to J20 mice treated with VX-765, arguing against the effects of VX-765 being solely due to inhibition of Casp1.

As indicated above, we have conducted a more thorough analysis of Iba1 in J20/Casp1^{+/+}, J20/Casp1^{+/-} and J20/Casp1^{-/-}. The data is shown as new Fig. 4c. The results show a replication of the effect seen with VX-765 treatment suggesting that VX-765 effect is occurring through Casp1.

7) Supplementary Figure 6 shows photomicrographs of Abeta staining in J20/Casp1^{+/+} and J20/Casp1^{-/-} mice, but the relative levels of deposition are hard to gauge. Plaque load measurements would help differentiate background staining from plaque deposition.

We agree and have conducted A β analyses on the additional J20/Casp1^{+/+}, J20/Casp1^{+/-} and J20/Casp1^{-/-}. Quantitative measures of histological and ELISA A β subtypes are in Fig. 5c, and 5h-k.

8) Some data in the current paper may contradict the conceptual model presented in Supplementary Figure 7. In principle, the inflammasome pathway (Nlrp1 \rightarrow Casp1 \rightarrow Casp6) is blocked in Casp1^{-/-} mice. So, if inflammation drives plaque formation, then the J20/Casp1^{-/-} mice should not have plaques. However, this is not the case, as illustrated in Supplementary

Figure 6. One wonders if VX-765 has pleotropic effects – acting not only on Casp1 (and possibly other caspases) but also on APP processing and/or Abeta clearance (Figures 4h and 4i notwithstanding, as the groups sizes are rather small). This situation would not detract from the description of the effects of VX-765 but has implications in understanding how the treatment works.

As noted above, the number of cases studied for the Casp1 analyses were originally low, thus providing only subjective data. More rigorous analysis is included in the revised paper. The new data shows that the absence of Casp1 decreases the immunopositive A β in histological sections (Fig. 5c), consistent with the decrease in VX-765-treated hippocampus and cortex (Fig. 5b). These data therefore support our conceptual model in Supplementary Fig. 7 (new Supplementary Fig. 10).

Furthermore, it would not be expected that plaques disappear completely since other inflammation mechanisms have not been abrogated by VX-765 like the NF κ B pathway.

Nevertheless, the reviewer correctly identified a discrepancy. With these new experiments, we realized that the cremophor vehicle injection does exacerbate the inflammation and A β levels compared to untreated mice. However, this is returned to normal with VX-765 treatment.

We have added additional samples and no significant change in IDE or NEP mRNA or protein (Supplementary Figure 9) can explain a change in the A β levels.

Overall, this paper describes beautifully performed experiments showing remarkable effects of VX-765 on behavior, inflammation and amyloid deposition in J20 mice. The validity of the conclusions would be strengthened, however, by addressing the issues delineated above or by modifying the conclusions.

Reviewer #3 (Remarks to the Author):

In this manuscript, Flores et al. show in vivo proof of concept for use of VX-765, a caspase-1 inhibitor, as a treatment for Alzheimer disease (AD) in the J20 familial AD mouse model. They present thorough in vitro selectivity data to demonstrate that VX-765 selectively inhibits caspase-1 in human and mouse and demonstrate that this compound rescues cognitive and memory deficits and attenuates hyperactivity in the J20 mouse model. These effects are shown to be dependent on VX-765 dose, and their dependence on caspase-1 inhibition was demonstrated in a caspase-1 null mouse. Further, VX-765 was shown to reduce markers of neuroinflammation, to inhibit the accumulation of A β , and to restore levels of synaptophysin. In human primary neuron cultures, VX-765 was demonstrated to prevent human neuronal degeneration on the basis of its ability to inhibit neuritic beading. Taken together, these results represent a convincing in vivo proof of concept for caspase-1 inhibition by VX-765 as a treatment for AD. Based on the absence of treatments that delay or reverse cognitive deficits associated with AD, this manuscript represents a significant advance and is well suited for publication in Nature Communications with minor revisions.

Comments:

1. It is interesting that VX-765 attenuates hyperactivity in J20 mice, but that hyperactivity persists in the caspase-1 null mouse. Can the authors comment on this?

By examining a high number of mice, it was observed that J20 lacking Casp1 (n=10-12) have not lost their hyperactivity. However, this cohort moves a lot less than the J20 cohort suggesting that the additional manipulation (injections and repeated behavioral studies) exacerbate J20 hyperactivity.

2. In general, the accessibility of the paper to a general audience could be improved. For example, the significance of thioflavin S and Iba1 staining could be explained briefly in the results section, and more explanation of which hypothesis each experiment is designed to test could be given.

We have revised the manuscript to explain more clearly our hypotheses and to make the paper easier to follow.

REVIEWERS' COMMENTS:

Reviewer #3 (Remarks to the Author):

In this revised manuscript, the authors the thoroughly addressed all of my comments from the first round of review, as well as comments from the other reviewer. This has substantially strengthened an already interesting and thorough manuscript, which is now well suited for publication in Nature Communications without further revision.